# Stress-responsive FKBP51 regulates AKT2-AS160 signaling and metabolic function

Georgia Balsevich[1], Alexander S. Häusl[1], Carola W. Meyer[2], Stoyo Karamihalev[1], Xixi Feng[3], Max L. Pöhlmann[1], Carine Dournes[1], Andres Uribe-Marino[1], Sara Santarelli[1], Christiana Labermaier[1], Kathrin Hafner[3], Tianqi Mao[3], Michaela Breitsamer[4], Marily Theodoropoulou[3], Christian Namendorf[3], Manfred Uhr[3], Marcelo Paez-Pereda[3], Gerhard Winter[4], Felix Hausch[5], Alon Chen[1], Matthias H. Tschöp[2], Theo Rein [3], Nils C. Gassen[3] & Mathias V. Schmidt [1]

The co-chaperone FKBP5 is a stress-responsive protein-regulating stress reactivity, and its genetic variants are associated with T2D related traits and other stress-related disorders. Here we show that FKBP51 plays a role in energy and glucose homeostasis. Fkbp5 knockout (51KO) mice are protected from high-fat diet-induced weight gain, show improved glucose tolerance and increased insulin signaling in skeletal muscle. Chronic treatment with a novel FKBP51 antagonist, SAFit2, recapitulates the effects of FKBP51 deletion on both body weight regulation and glucose tolerance. Using shorter SAFit2 treatment, we show that glucose tolerance improvement precedes the reduction in body weight. Mechanistically, we identify a novel association between FKBP51 and AS160, a substrate of AKT2 that is involved in glucose uptake. FKBP51 antagonism increases the phosphorylation of AS160, increases glucose transporter 4 expression at the plasma membrane, and ultimately enhances glucose uptake in skeletal myotubes. We propose FKBP51 as a mediator between stress and T2D development, and potential target for therapeutic approaches.

[1] Department of Stress Neurobiology and Neurogenetics, Max Planck Institute of Psychiatry, Kraepelinstraße 2-10, 80804 Munich, Germany. [2] Institute of Diabetes and Obesity, Helmholtz Zentrum München, Parkring 13, 85748 Garching, Germany. [3] Department of Translational Research in Psychiatry, Max Planck Institute of Psychiatry, Kraepelinstraße 2-10, 80804 Munich, Germany. [4] Ludwig Maximilians University, Butenandtstr. 5-13, 81377 Munich, Germany. [5] Technical University Darmstadt, Institute of Organic Chemistry and Biochemistry, Alarich-Weiss-Str. 4, 64287 Darmstadt, Germany. Georgia Balsevich and Alexander S. Häusl contributed equally to this work. Theo Rein, Nils C. Gassen and Mathias V. Schmidt jointly supervised this work. Correspondence and requests for materials should be addressed to G.B. (email: georgia.balsevich@ucalgary.ca) or to M.V.S. (email: mschmidt@psych.mpg.de)

FK506-binding protein 51 (FKBP51) is an immunophilin protein best known as a negative regulator of the glucocorticoid receptor (GR) and consequently the physiological stress response[1]. Specifically, in complex with FKBP51 (*FKBP5* gene), the GR displays reduced ligand affinity, reduced nuclear translocation, and ultimately decreased GR sensitivity[1–5]. By contrast, in complex with its functional counter-player FKBP52 (*FKBP4* gene), GR activity is enhanced[6]. Single-nucleotide polymorphisms (SNPs) in the *FKBP5* gene, which are associated with increased expression of *FKBP5* (high-induction allele), have been fundamentally linked to stress-related disorders, and most notably in psychiatric disorders[7]. In this context, it is possible that stress-induced *FKBP5* may similarly be implicated in additional stress-related pathophysiologies, such as type 2 diabetes (T2D).

Exposure to nutrient overload, including exposure to a high-fat diet (HFD), is considered a metabolic stressor[8]. Interestingly, it was reported that 8 weeks of HFD exposure in mice led to enhanced *Fkbp5* expression in the hypothalamus[9], suggesting that *Fkbp5* is responsive to metabolic stressors and is able to sense the nutrient environment. Accordingly, a study examining food restricted-responsive genes reported an induction of *Fkbp5* in the hypothalamus and ventral tegmental area[10]. This is in agreement with an earlier study, which used a 24-h food restriction paradigm as a stressor to investigate stress-induced *Fkbp5* expression across multiple brain regions[11].

There are several additional lines of evidence to support the possibility that FKBP51 links stress to metabolic function. Human adipocytes and skeletal muscle are among the tissues presenting the strongest expression of *FKBP5*[12]. Recently, the high-induction *FKBP5* risk allele was associated with reduced weight loss following bariatric surgery[13]. A genome-wide association study furthermore demonstrated that SNPs within the *FKBP5* gene loci are associated with T2D and markers of insulin resistance[14]. Preclinical studies in animal models have similarly demonstrated that complete loss of FKBP51 protects against HFD-induced body weight gain and hepatic steatosis, which is, in part, explained by an increased expression of uncoupling protein 1 (UCP1), a specific marker of browning, in white adipose tissue (WAT) and increased thermogenesis[15]. Additionally, FKBP51 is a negative regulator of all 3 isoforms of the serine/threonine protein kinase AKT (AKT1, AKT2, and AKT3), and through this action, regulates the response to chemotherapy[16]. AKT is also a central node within the insulin signaling pathway, and deregulation of AKT activation, most notably AKT2 activation, has been linked to the pathogenesis of diabetes and obesity[17, 18]. In this context, FKBP51 may be an important regulator of insulin signaling and consequently energy and glucose homeostasis[19]. Nevertheless, whether FKBP51 plays a critical role in whole body glucose metabolism remains to be elucidated. For this purpose, we aimed to characterize the role of FKBP51 in energy and glucose homeostasis using a combination of *Fkbp5* knockout (51KO) mice,

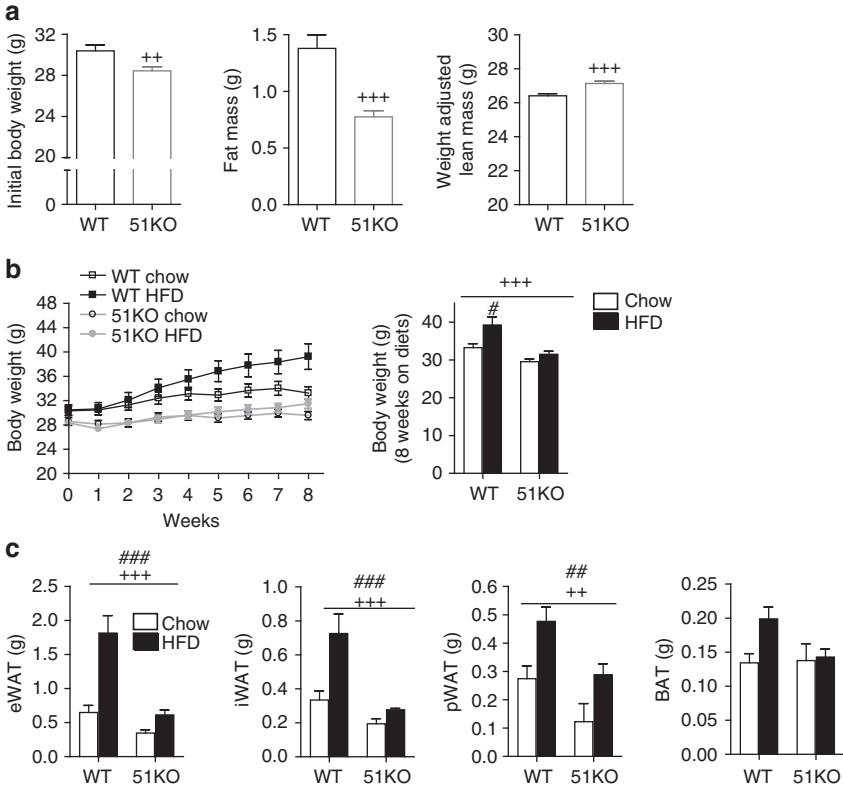

**Fig. 1** Genetic ablation of FKBP51 prevents HFD-induced weight gain. **a** 51KO mice ($n = 16$) presented lowered body weight, decreased fat mass, and increased lean mass compared to WT littermates ($n = 18$) at the onset of the dietary feeding period. **b** 51KO mice weighed significantly less than WT mice throughout the 8-week dietary treatment and at the experimental end ($n = 9$ WT-Chow, $n = 9$ WT-HFD, $n = 9$ 51KO-Chow, $n = 7$ 51KO-HFD). Whereas WT mice were susceptible to HFD-induced weight gain, 51KO mice were not as interpreted from weight progression and final body weight (following 8 weeks on respective diets). **c** After 8 weeks on the dietary treatment, 51KO mice presented decreased fat pad weights for epididymal (e), inguinal (i), and perirenal (p) white adipose tissues (WAT) compared to WT counterparts whereas presented no change in brown adipose tissue (BAT) mass. HFD exposure significantly increased fat pad mass, regardless of genotype. Lean mass was adjusted for body weight and is expressed for a 30-g mouse. Data are represented as mean ± SEM. $^{+}P < 0.05$, $^{++}P < 0.01$, $^{+++}P < 0.001$; $^{\#}P < 0.05$, $^{\#\#}P < 0.01$, $^{\#\#\#}P < 0.001$, two-tailed $t$ test for **a**, Repeated measures ANOVA and two-way ANOVA for **b**, two-way ANOVA for **c**; + significant genotype effect; # significant diet effect

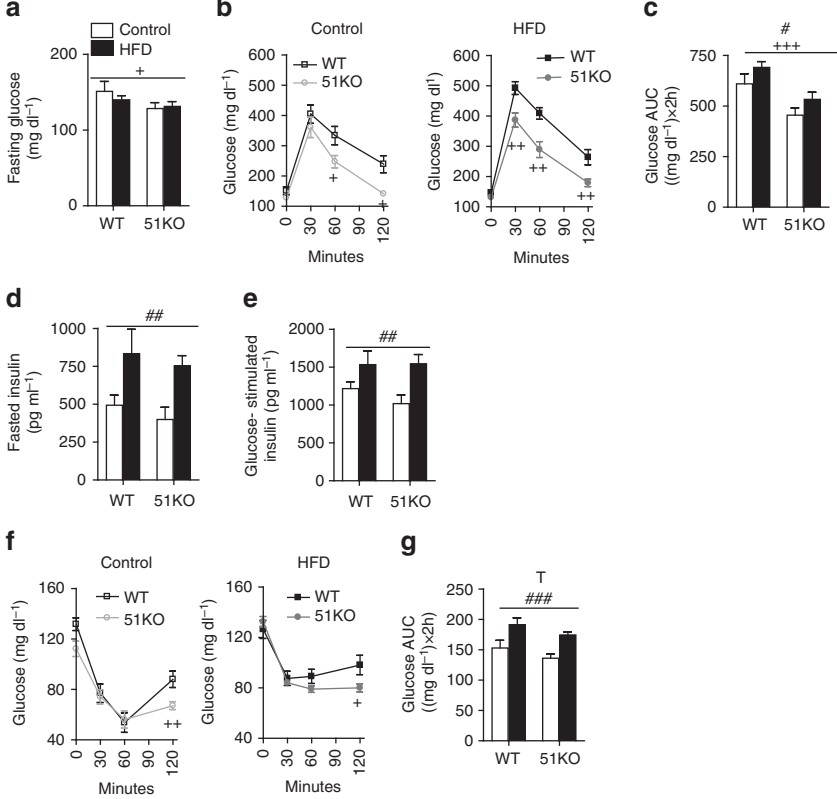

**Fig. 2** Genetic ablation of FKBP51 improves glucose tolerance. **a** Blood glucose following a 14 h fast was significantly lower in 51KO mice compared to WT mice. **b** In the GTT, a HFD impaired glucose tolerance in WT mice but not in 51KO mice. **c** The glucose area under curve (AUC) illustrates the effect of genotype and diet on glucose tolerance. **d**, **e** Fasted insulin and glucose-stimulated insulin were significantly elevated from HFD exposure independent of genotype. **f** In HFD-fed mice, loss of FKBP51 significantly reduced insulin tolerance. Importantly blood glucose remained significantly lower 120 min following insulin administration on account of FKBP51 deletion under both chow conditions and HFD conditions. **g** The glucose AUC for ITT demonstrates the strong diet effect and a trend for genotype. $n = 8$ WT-Control, $n = 10$ WT-HFD, $n = 12$ 51KO-Control, $n = 13$ 51KO-HFD. Data are represented as mean ± SEM. $^{+}P < 0.05$, $^{++}P < 0.01$, $^{+++}P < 0.001$; $^{\#}P < 0.05$, $^{\#\#}P < 0.01$, $^{\#\#\#}P < 0.001$, $^{T}P < 0.1$, two-way ANOVA for **a**, Repeated measures ANOVA for **b**, **f**, two-way ANOVA for **c**–**e** and **g**; + significant genotype effect; # significant diet effect; T significant trend for genotype

pharmacological manipulations, and mechanistic studies. We found in this study that FKBP51 regulates glucose metabolism in mice, through it, regulation of AKT2-AS160 signaling, glucose transporter expression, and glucose uptake in myotubes. Pharmacological antagonism of FKBP51 improves glucose tolerance, irrespective of body weight changes, which suggests an opportunity to target FKBP51 for the treatment of T2D.

## Results

**FKBP51 loss opposes obesity and improves glucose tolerance**. In order to examine the role of FKBP51 in energy and glucose homeostasis, we initially characterized the metabolic outcomes arising in 51KO mice. We found that 51KO mice fed with a standard chow diet showed a modest body weight reduction, reduced adiposity, and increased lean mass compared to WT littermates (Fig. 1a). When challenged with HFD exposure for 8 weeks, the 51KO mice were protected from both HFD-induced weight gain and increased adiposity (Fig. 1b, c). Loss of FKBP51 likewise counteracted diet-induced obesity under thermoneutral conditions (30 °C), arguing against a thermoregulatory basis of the phenotype (Supplementary Fig. 1). Indirect calorimetry revealed that the body weight phenotype observed in 51KO mice under standard chow conditions was accompanied by a modest increase in total energy expenditure, as a result of an increased resting metabolic rate (RMR) (Supplementary Fig. 2A). In

addition, 51KO mice presented a modest decrease in their respiratory exchange ratio (RER) and a slight increase in their home-cage activity (Supplementary Fig. 2B, C). By contrast, neither water nor food intakes were affected by loss of FKBP51 (Supplementary Fig. 2D, E). To confirm a lack of FKBP51 effect on feeding behavior, a separate pair-feeding experiment was performed, in which a cohort of WT mice was pair-fed to 51KO mice. This experiment again revealed no genotype effect on energy intake (Supplementary Fig. 2G). Cold-induced body temperature regulation was unaffected by FKBP51 genotype (Supplementary Fig. 2H).

To determine the effects of FKBP51 on glucose metabolism and insulin sensitivity, we performed glucose tolerance tests (GTTs) and insulin tolerance tests (ITTs) in a separate cohort of 51KO and WT mice. The body weight data were consistent with our previous experiments (Supplementary Fig. 3). FKBP51 deletion lowered fasting glucose (Fig. 2a) and remarkably improved glucose tolerance (Fig. 2b, c). Although the effect of FKBP51 deletion was present under the control condition, a metabolic challenge using HFD exposure heightened the genotype effect. Interestingly, the levels of fasted insulin and glucose-stimulated insulin were not different between 51KO and WT mice (Fig. 2d, e), indicating that differences in insulin secretion do not contribute to the improved glucose tolerance phenotype. In the ITT, 51KO mice presented a prolonged response to insulin under control and HFD conditions, despite

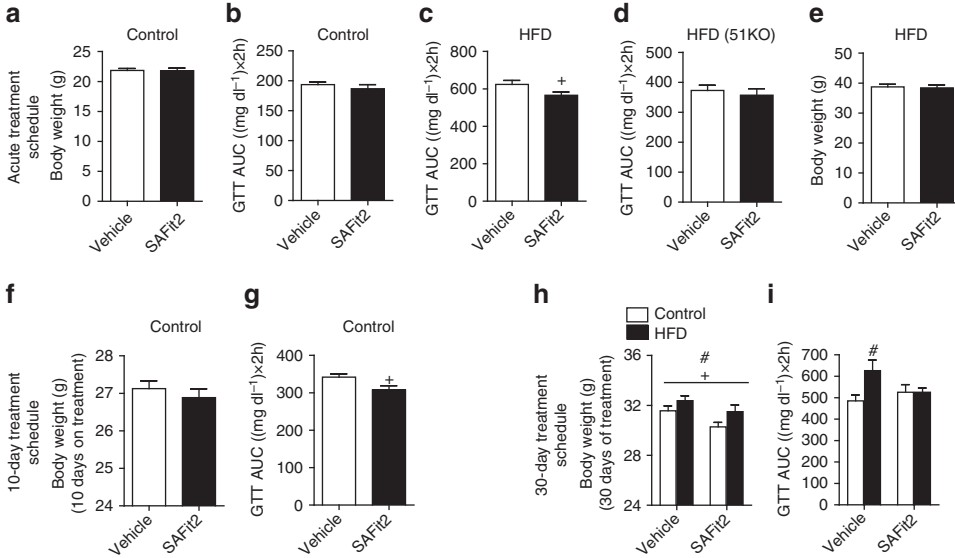

**Fig. 3** FKBP51 antagonism parallels the metabolic effects resulting from genetic ablation of FKBP51. **a** A single application of a slow-release-formulated SAFit2 gel had no effect on glucose tolerance or (**b**) Body weight under control diet conditions. **c** Under HFD conditions, acute administration of SAFit2 gel significantly improved glucose tolerance. **d** The effects of SAFit2 on glucose tolerance under HFD conditions were not present in 51KO. **e** Despite the effects of acute SAFit2 on glucose tolerance under HFD conditions, there was no effect on body weight (**f**) 10-day SAFit2 treatment had no significant effect on body weight. **g** Despite no effect on body weight, SAFit2 treatment significantly improved glucose tolerance as reflected in the glucose area under the curve (AUC) for the GTT measured on treatment day 8. **h** At the experimental end point (following 30 days of treatment), mice treated with SAFit2 weighed significantly less than their diet counterparts. Nevertheless, mice fed with the HFD remained significantly heavier independent of treatment. **i** The extended SAFit2 treatment schedule furthermore protected against HFD-induced impaired glucose tolerance as reflected in the glucose AUC measured on day 25. For acute treatment schedule in C57BL6 $n = 12$ per treatment group; for acute treatment in 51KO $n = 8$ per treatment group. For the 10-day treatment schedule, $n = 8$ per treatment group. For 30-day treatment schedule $n = 12$ Vehicle-Control, $n = 13$ SAFit2-Control, $n = 12$ Vehicle-HFD, $n = 13$ SAFit2-HFD. The data are represented as mean ± SEM. $^{+}P < 0.05$; $^{\#}P < 0.05$, two-tailed $t$ test for **a**–**g**, two-way ANOVA for **h**, two-way ANOVA plus Bonferroni testing for **i**; + significant treatment effect; # significant diet effect

the fact that both 51KO and WT mice remained vulnerable to HFD-induced insulin intolerance (Fig. 2f, g).

**FKBP51 antagonism rapidly improves glucose tolerance**. Due to the improved metabolic phenotype presented in 51KO mice, we subsequently assessed the effects of pharmacological blockade of FKBP51. Importantly, the first highly selective antagonist for FKBP51, SAFit2, has recently been developed[20]. In the current study, we determined the efficacy of FKBP51 antagonism in mice on metabolic parameters. We examined the efficacy of a single application of SAFit2 gel (a slow release formulation, Supplementary Fig. 4A), to improve glucose tolerance. Although SAFit2 gel had no effect in chow-fed mice measured at 48 h following application (Fig. 3a, b), it significantly improved glucose tolerance in HFD-fed mice (Fig. 3c), supporting the literature demonstrating that FKBP51-mediated outcomes are more robust under challenging conditions[21–25]. The effect of SAFit2 treatment on glucose tolerance is specific to FKBP51 inhibition, as no effect was observed in 51KO mice under HFD conditions (Fig. 3d). Importantly, body weight was not affected (Fig. 3e), arguing for a body weight-independent effect of FKBP51 inhibition on glucose tolerance. Acute SAFit2 treatment furthermore had no effect on energy expenditure, respiratory exchange ratio, or activity counts (Supplementary Fig. 4B–D). We also examined the expression of UCP1 in brown adipose tissue (BAT) of mice treated for 48 h with SAFit2 and found no effect of acute treatment (Supplementary Fig. 4E). We then examined the effects of a sub-chronic SAFit2 (20 mg kg$^{-1}$) regimen administered twice daily for 10 days to adult C57BL/6 mice by intraperitoneal injections under chow-fed conditions. The dose was selected based on the effective dose used in acute studies[26]. Using a 10-day treatment schedule, we observed an effect of SAFit2 under chow diet conditions.

Specifically, although 10 days of FKBP51 antagonism again yielded no body weight phenotype, there was a marked improvement of glucose tolerance assessed on treatment day 8 (Fig. 3e, f). The lack of body weight change from either the acute or 10-day SAFit2 treatment schedule indicates that the FKBP51-dependent glucose tolerance phenotype is not secondary to the body weight phenotype. To determine whether a longer treatment period would replicate the body weight phenotype of 51KO mice, a new cohort of C57BL/6 mice was treated with SAFit2 for 30 days under both control and HFD conditions (for SAFit2 plasma levels see Supplementary Fig. 5A). At the onset of treatment, following 4 weeks of dietary exposure, there was no difference in body weight between the treatment groups (Supplementary Fig. 5B, C). However, we found that 30 days of SAFit2 administration led to a reduction in body weight under both control and HFD conditions (Fig. 3g). FKBP51 antagonism furthermore protected against HFD-mediated glucose intolerance (Fig. 3h). There was, however, no effect of SAFit2 on insulin tolerance or on locomotor activity tested in the open-field test (Supplementary Fig. 5D, E). We observed no unwanted side-effects of FKBP51 antagonism on behavioral readouts tested in the dark–light transition and elevated plus-maze tests (Supplementary Fig. 5F, G). Taken together, these results clearly demonstrate that (i) a body weight phenotype is secondary and not necessary for the effects of FKBP51 antagonism on glucose tolerance, and (ii) pharmacological blockade of FKBP51 parallels the effects of FKBP51 genetic ablation.

**FKBP51 loss sensitizes insulin signaling in skeletal muscle**. As the effect of FKBP51 on glucose tolerance was primary and independent to the body weight phenotype, we addressed this mechanism in the subsequent experiments. To unravel the FKBP51-dependent regulation of glucose metabolism, we first

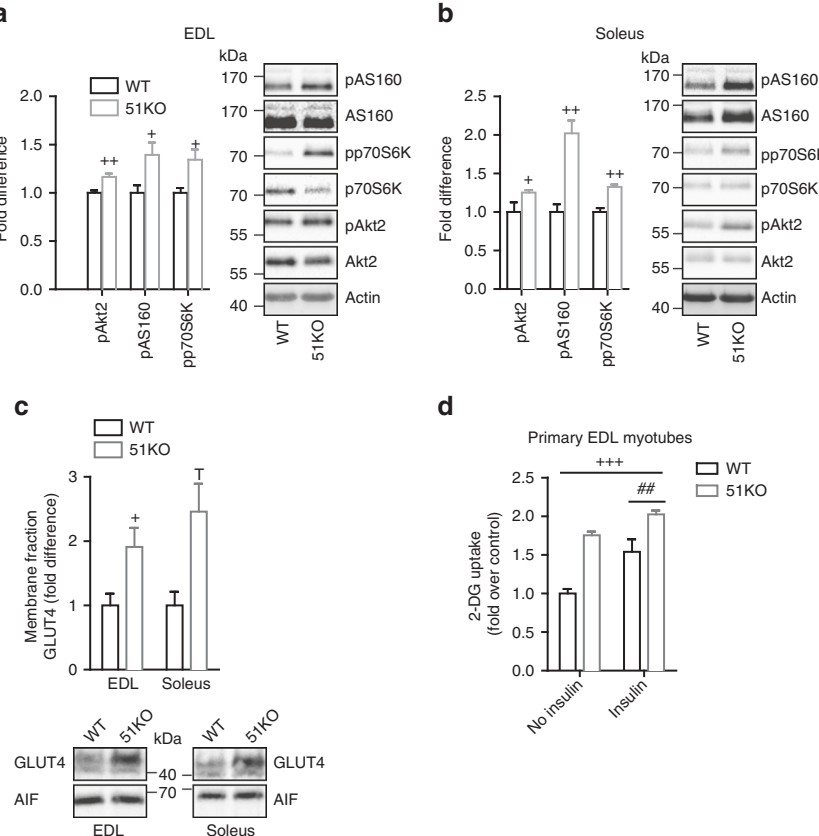

**Fig. 4** FKBP51 affects insulin signaling and consequently glucose uptake. **a**, **b** Insulin signaling was enhanced in EDL (**a**) and soleus (**b**) skeletal muscles of 51KO mice compared to WT mice as assessed by pAKT2, pAS160, and pp70S6K protein expression. **c** Following subcellular fractionation to isolate the plasma membrane compartment, we observed increased GLUT4 expression in skeletal muscle membrane fractions of 51KO mice. **d** In primary EDL myotubes, loss of FKBP51 heightened glucose uptake under both no insulin and insulin-stimulated states. For quantification of phosphorylated protein, $n = 6$ per group. For GLUT4 membrane localization, $n = 3$ per group. For glucose uptake experiments, 3 wells for each condition were measured. The data are expressed as relative fold change compared to wild-type condition ± SEM. $^+P < 0.05$, $^{++}P < 0.01$, $^{+++}P < 0.001$, $^{##}P < 0.01$, two-tailed $t$ tests for **a–d**, two-way ANOVA for **e**; + significant genotype effect, # significant insulin effect, T trend for genotype. Supplementary Fig. 12 shows uncropped gel images

examined FKBP51 protein expression across multiple peripheral tissues. Interestingly, FKBP51 was not ubiquitously expressed across all tissues examined, but rather showed a defined expression profile. FKBP51 was detected within skeletal muscle (soleus muscle and EDL) eWAT and iWAT (Supplementary Fig. 6A). By contrast, FKBP51 was not detected in the liver, kidney, spleen, pancreas, gut, or BAT. In addition, HFD exposure (for 8 weeks) significantly increased levels of FKBP51 in EDL skeletal muscle (Supplementary Fig. 6B), and supports the notion that a high-fat dietary environment acts as a metabolic stressor[8, 27].

As a next step, we examined the phosphorylation status of critical nodes along the insulin signaling cascade as a marker of insulin signaling activation. As we found a strong effect of FKBP51 loss on glucose tolerance without any change in circulating insulin levels, we hypothesized that intracellular insulin signaling is enhanced following loss of FKBP51. Indeed, 5 min following insulin stimulation ($0.70\,\text{IU kg}^{-1}$), we found that insulin sensitivity (as reflected through the phosphorylation status of AKT2, AS160, and p70S6K) was markedly increased in skeletal muscle (soleus muscle and EDL) of 51KO mice (Fig. 4a, b), whereas insulin activation in WAT (iWAT and eWAT) and liver remained unchanged (Supplementary Fig. 7A, C). The skeletal muscle-specific effect of FKBP51 deletion on insulin signaling is in line with the high expression level of FKBP51 across skeletal muscle tissues (reported above).

We subsequently assessed whether events downstream of AKT2-AS160 signaling are likewise regulated by FKBP51.

Therefore, we first examined glucose transporter GLUT4 translocation to the plasma membrane, which is triggered by activated AKT2-AS160 signaling[28]. We isolated the plasma membrane fraction from skeletal muscle extracted from WT and 51KO mice. 51KO mice display increased GLUT4 expression compared to WT mice (Fig. 4c). There was no difference in GLUT1 expression in the plasma membrane fraction between WT and 51KO mice. Importantly, there was no effect of FKBP51 deletion either on total GLUT1 or GLUT4 expression (Supplementary Fig. 7D). To assess the functional implications of these FKBP51-dependent events, we examined radiolabeled-2-deoxyglucose uptake in primary EDL myotubes collected either from WT or 51KO mice. Glucose uptake was significantly increased by both insulin and FKBP51 deletion (Fig. 4d). Moreover, the effect of FKBP51 ablation on glucose uptake was observed under both non-insulin and insulin-stimulated states, indicating that FKBP51 deletion enhances glucose uptake independent of the insulin state. In support of these findings, overexpression of AKT2 enhanced glucose uptake in C2C12 myotubes (a mouse skeletal muscle cell line), whereas simultaneous overexpression with FKBP51 prevented the AKT2-mediated enhanced uptake (Supplementary Fig. 8). Thus, FKBP51 acts along the AKT2-AS160 signaling pathway to dampen glucose uptake.

Given the effects of the FKBP51 antagonist, SAFit2, on glucose tolerance in mice, we sought to further investigate whether SAFit2 modulates AKT2 activation and downstream events based on the

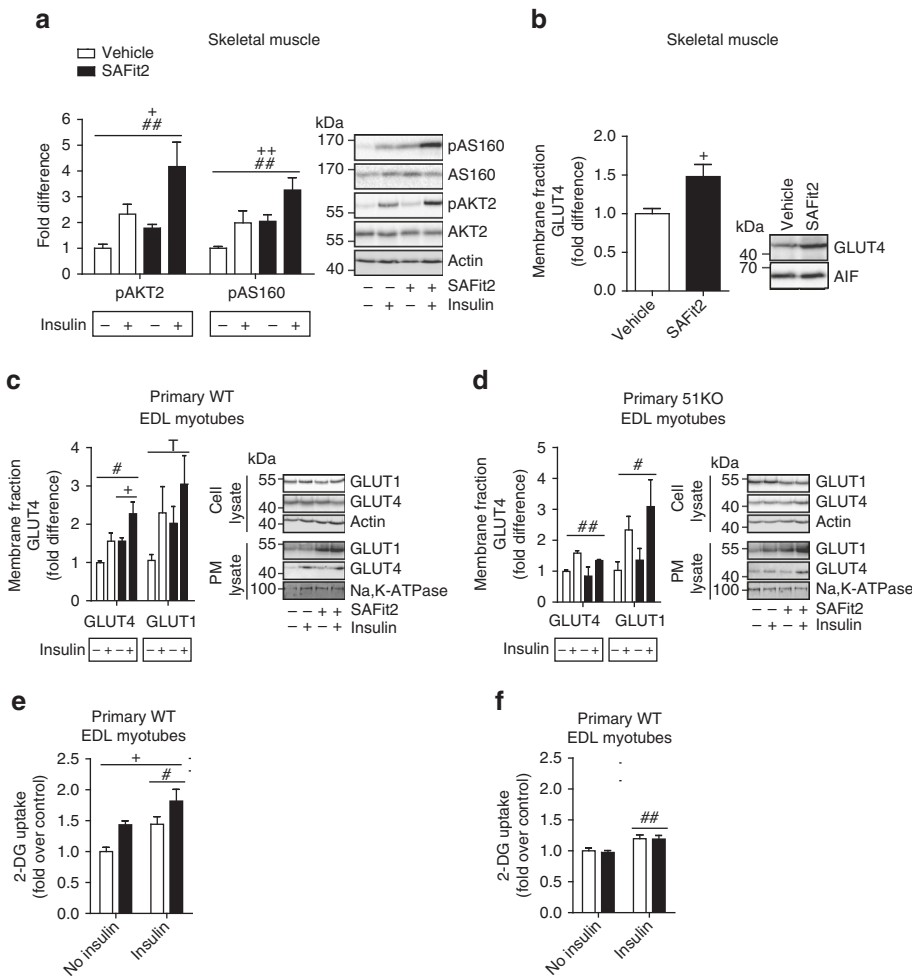

**Fig. 5** FKBP51 antagonism affects insulin signaling and consequently glucose uptake. **a** The insulin signaling pathway was enhanced in EDL skeletal muscle of mice treated with SAFit2 compared to vehicle-treated mice, independent of insulin, as assessed by pAKT2, and pAS160 protein expression. **b** GLUT4 expression at the membrane was increased 6 h following SAFit2 treatment in soleus skeletal muscle. **c** GLUT4 expression at the membrane was increased from SAFit2 treatment in primary EDL myotubes from WT mice, whereas GLUT1 expression was unchanged by SAFit2 treatment. **d** SAFit2 had no effect on GLUT4 plasma membrane expression in primary EDL muscle cells collected from 51KO mice. **e** FKBP51 antagonism with SAFit2 increased 2-deoxyglucose uptake in primary EDL muscle cells collected from WT mice independent of insulin condition. **f** SAFit2 had no effect on 2-deoxyglucose uptake in primary 51KO muscle cells. For quantification of phosphorylated protein expression in mice, $n = 6$ per group. For quantification of GLUT4 expression in mice, $n = 7$ per treatment. For GLUT1/4 expression in primary EDL myotubes, $n = 3$ per group. For glucose uptake experiments, 3 wells for each condition were measured. Data are expressed as relative fold change compared to vehicle condition $\pm$ SEM. $^+P < 0.05$, $^{++}P < 0.01$, $^\#P < 0.05$, $^{\#\#}P < 0.01$, two-way ANOVA for **a**–**f**; + significant treatment effect, # significant insulin effect, T trend ($p < 0.1$) for insulin effect. Supplementary Figs. 13 and 14 show uncropped gel images

known regulation of AKT2 by FKBP51[16]. We assessed pAKT2 and pAS160 in tissue collected from mice that had been pre-treated with vehicle or SAFit2 6 h prior to tissue collection in either an insulin-stimulated ($0.70\ \mathrm{IU\,kg^{-1}}$ 5 min before tissue collection) or non-insulin (saline) condition. Although a single intraperitoneal injection of SAFit2 applied 6 h before testing had no effect on glucose tolerance (Supplementary Fig. 9), there was already a significant effect of SAFit2 treatment at a molecular level. SAFit2 significantly increased phosphorylated AKT2 and AS160 in EDL muscle and likewise increased expression of GLUT4 at the membrane in soleus muscle (Fig. 5a, b). Once again, the effect of FKBP51 antagonism on pAKT2-pAS160 was independent of the insulin-stimulated state. The same directional change on phosphorylation states was observed in soleus muscle, whereas this was not seen in eWAT (where FKBP51 expression is relatively low) (Supplementary Fig. 10A, B). To ensure that the effects of SAFit2 were selective for FKBP51, we additionally examined the effects of SAFit2 treatment on pAKT2 and pAS160

in primary EDL and soleus myotubes from WT and 51KO mice. While SAFit2 treatment increased the expression of pAKT2 (soleus and EDL muscle) and pAS160 (EDL muscle) in WT cells, there was no effect of FKBP51 antagonism in 51KO cells, confirming the specificity of SAFit2 action (Supplementary Fig. 10C, F). Moreover, following SAFit2 treatment, GLUT4 expression increased in the membrane fraction of primary EDL myotubes from WT mice, but not from 51KO mice (Fig. 5c, d). Finally, we applied SAFit2 to primary EDL myotubes and measured radiolabeled-2-deoxyglucose uptake. FKBP51 antagonism-increased glucose uptake in EDL cells collected from WT mice, but not 51KO mice, again demonstrating the specificity of SAFit2 (Fig. 5e, f). Taken together, FKBP51 antagonism increases AKT2-AS160 signaling, GLUT4 expression at the plasma membrane, and glucose uptake into primary myotubes.

From these results, and based on previous literature high-lighting a role of FKBP51 as a scaffolding protein[16, 21, 29], we assessed whether FKBP51 forms a complex with AS160 and

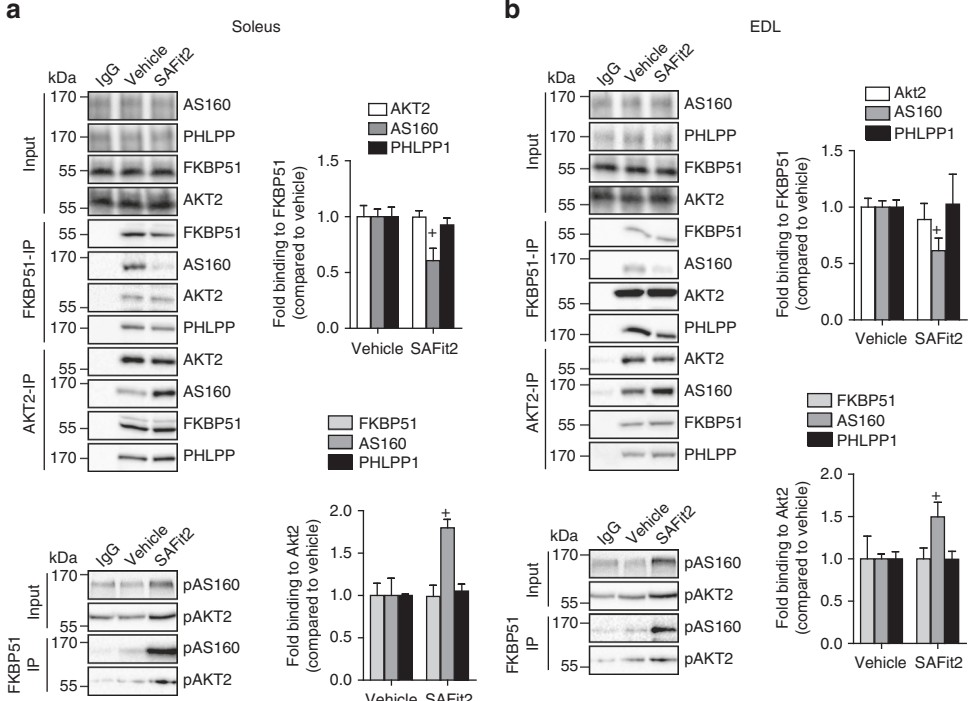

**Fig. 6** FKBP51 antagonism affects AKT2-AS160 signaling complex. Tissue lysates from 30-day vehicle-treated or SAFit2-treated mice exposed to HFD were immunoprecipitated with anti-AKT2 and anti-FKBP51 and then analyzed by Western blot using FKBP51, (p)AKT2, (p)AS160, and PHLPP1. **a**, **b** Immunoprecipitation reactions revealed that SAFit2 treatment increased binding between (p)AKT2 and (p)AS160 in soleus (**a**) and EDL (**b**) muscles, while simultaneously decreased binding between FKBP51 and AS160 in both muscle types. For co-immunoprecipitation experiments $n = 3$ per group. Data are expressed as relative fold change compared to vehicle condition $\pm$ SEM. $^{+}P < 0.05$, two-tailed $t$ tests for **a**, **b**; + significant SAFit2 treatment effect. Supplementary Fig. 15 shows uncropped gel images

furthermore assessed whether SAFit2 modulates this interaction. Immunoprecipitation reactions, using protein extracts from 30-day vehicle-treated and SAFit2-treated mice, revealed that SAFit2 treatment strengthened the binding between AKT2 and AS160 (Fig. 6a, b, bottom). This is in line with increased glucose uptake given that AKT2 inactivates AS160, which in turn stimulates the translocation of GLUT4 to the plasma membrane[28]. We furthermore observed a novel interaction between FKBP51 and AS160, and confirmed that SAFit2 disrupts this interaction (Fig. 6a, b, top). SAFit2, by contrast, had no effect on the binding strength between AKT2 and PHLPP1 (the negative regulator of AKT2). We confirmed that the phosphorylation state of AKT2 and AS160 does not interfere with the interaction between FKBP51 and either AKT2 or AS160 by also examining pAKT2 and pAS160. Co-immunoprecipitation does not discriminate between direct and indirect protein interactions, and it is possible that intermediate proteins are also involved in the interaction between FKBP51 and AS160. Regardless, these findings suggest that FKBP51 associates (directly or indirectly) with AS160 to reduce glucose uptake. By contrast, SAFit2 antagonizes FKBP51 and reduces the binding between FKBP51 and AS160 to promote a steric arrangement that favors glucose uptake.

We have shown that FKBP51 regulates insulin signaling selectively in skeletal muscle. Yet FKBP51 is nonetheless expressed in select WAT depots, albeit at lower levels. In order to account for the tissue-specific effects of FKBP51, we furthermore investigated the expression profile of FKBP52, a structural homolog of FKBP51 that competes as a scaffolding protein, resulting in opposite functional effects[30]. Interestingly, the expression profile between FKBP51 and FKBP52 across WATs and skeletal muscle are very distinct and provide an explanation for the FKBP51 skeletal muscle-selective effects.

Specifically, FKBP51 expression is relatively high in skeletal muscle compared to WATs (Supplementary Fig. 6A). In stark contrast, levels of FKBP52 are high in WAT depots relative to skeletal muscle. (Supplementary Fig. 11A). We additionally performed co-immunoprecipitation reactions with FKBP52 to investigate whether FKBP51 and FKBP52 compete for binding to AKT2. We demonstrate that similar to FKBP51, FKBP52 is in complex with AKT2 and AS160 (Supplementary Fig. 11B). However, FKBP52 does not likewise interact with PHLPP1 (Supplementary Fig. 11B), the negative regulator of AKT2, and consequently has divergent effects on downstream AKT2 signaling. Specifically, where ectopic overexpression of FKBP51 significantly decreased pAKT2 and pAS160 expression in C2C12 myotubes, simultaneous overexpression of FKBP52 abolished the effects of ectopic FKBP51 (Supplementary Fig. 10C). Taken together, FKBP51 and FKBP52 compete for binding with AKT2. The distinct expression profile and functional outcomes of FKBP51 and FKBP52 are responsible for the skeletal muscle-specific effects of SAFit2, which selectively antagonizes FKBP51.

## Discussion

Here we describe that loss of FKBP51 in mice markedly improves metabolism and especially improves glucose tolerance under both control and HFD conditions. This is in line with earlier pre-clinical and human studies, which identified an association between FKBP51 ablation and *FKBP5* SNPs on traits related to body weight regulation and T2D, respectively[14, 15]. Although the effects of FKBP51 loss are witnessed under dietary control conditions, the effects are significantly accentuated when mice are challenged to an HFD, which acts as a metabolic stressor[8]. This supports a large body of literature, demonstrating that an

environmental challenge is a prerequisite for FKBP51-mediated outcomes. For example, early life trauma (i.e., environmental challenge) increases the risk of various psychiatric disorders selectively in *FKBP5* risk allele carriers, which are associated with increased FKBP51 protein levels[7, 24, 25, 31–33]. In rodent studies, 51KO mice present no overt phenotype under basal conditions, yet show improved stress resilience following either acute stress[23] or chronic stress[22]. Indeed, stressors induce *FKBP5* expression, and this may underlie the more pronounced effects of FKBP51 deletion on metabolic phenotypes seen in the current study. We certainly found that an HFD increases levels of FKBP51 in fat and skeletal muscle. Taken together, previous findings have found that higher levels of FKBP51 are associated with poorer outcomes in stress-related psychiatric disorders. The present study shows that higher levels of FKBP51 are likewise detrimental to metabolic health, especially when confronted with environmental challenges (i.e., an obesogenic environment).

Selective pharmacological antagonism of FKBP51 has only recently been realized[20]. Selectivity for FKBP51 is especially important since its structural homolog, FKBP52, acts as a functional opponent. In fact, it was this structural similarity that initially hampered drug discovery for FKBP51. The current ability to antagonize FKBP51 offers new opportunities for drug development. To date, only three studies have investigated the effects of FKBP51 antagonism on functions related to FKBP51, and no study has yet been investigated for long-term (i.e., 30 day) applicability. These previous studies independently found that FKBP51 antagonism induces anxiolytic effects[26], reduces the severity of pain symptoms[34], and opposes the known ability of FKBP51 to promote NFκB signaling[35]. Nevertheless, FKBP51 is a multi-domain protein[36], and it remains unknown whether the FKBP51 antagonist, SAFit2, blocks all functions of FKBP51. Therefore, in order to address whether pharmacological antagonism affects metabolic function, mice were treated with SAFit2 once (slow release formula) or repeatedly for either 10 or 30 days. Administration of SAFit2 paralleled the metabolic phenotype arising from total genetic loss of FKBP51. Acute SAFit2 treatment improved glucose tolerance under metabolically challenging conditions (i.e., HFD conditions). Under metabolic control conditions (i.e., regular diet), FKBP51 blockade improved glucose tolerance as early as 8 days following treatment onset. Importantly, the effects of FKBP51 modulation on glucose tolerance were not secondary to changes in body weight since neither a single nor a 10-day SAFit2 exposure had an effect on body weight. It is possible that our study was underpowered to detect the modest effects of SAFit2 treatment on body weight, since the 10-day SAFit2-treated group showed a lower (non-significant) body weight phenotype compared to the vehicle-treated counterparts. Regardless, these data support our studies in 51KO mice, which collectively indicate that FKBP51 is an integral component of glucose homeostatic regulation, particularly in response to nutritional changes. In the context of body weight regulation, our findings that 30 days of SAFit2 treatment protects against HFD-induced weight gain supports the findings of a recently published paper demonstrating that 51KO mice resist HFD-induced weight gain and present increased UCP1 in WAT[15]. One limitation of our study is that we do not know the minimal effective dose of SAFit2 required to improve glucose tolerance. Nevertheless, the improved metabolic outcomes following systemic administration of the selective FKBP51 antagonist clearly demonstrate the applicability in a clinical setting.

It is well established that FKBP51 is able to regulate many signaling pathways through direct protein–protein interactions[16, 21, 37, 38]. Through these interactions, FKBP51 has been implicated in various disease states (i.e., cancers, psychiatric disorders) and in the response to medications (i.e., chemotherapies,

antidepressants). For example, through the FKBP51-dependent regulation of AKT, Pei et al. (2009) reported that FKBP51 reduces tumor growth[16]. FKBP51 binds to both AKT1 and AKT2 isoforms and their corresponding negative regulators PHLPP2 and PHLPP1, to ultimately favor Akt inactivation. Interestingly, while Akt1 is best known for its regulation of cell growth[39, 40], AKT2 is best known for its regulation of glucose homeostasis[17, 41]. Accordingly, in the present study, we determined that FKBP51 is also important for glucose disposal through the regulation of AKT2 and downstream AS160 (AKT substrate of 160 kDa), an important signaling protein involved in insulin-stimulated glucose transport in skeletal muscle[28, 42]. 51KO mice exhibit enhanced insulin signaling, as interpreted from the increased phosphorylation of AKT2, AS160, and p70S6K. Interestingly, the FKBP51-dependent effects on insulin signaling observed in mice were highly tissue-specific, in which FKBP51-dependent increases in insulin signaling was limited to skeletal muscle. Indeed, skeletal muscle accounts for an estimated 80% of postprandial glucose disposal and is regarded as a principal site responsible for the maintenance of glucose homeostasis[43, 44]. In this context, we also found that primary EDL myotubes from 51KO mice exhibited heightened glucose uptake compared to WT EDL myotubes. Ectopic FKBP51 overexpression furthermore completely reversed the enhanced glucose uptake arising from AKT2 overexpression in cultured myotubes, demonstrating that FKBP51 regulation of insulin signaling is critical for glucose uptake.

The FKBP51-dependent effects on the phosphorylation of AKT2 and AS160 as well as on glucose uptake were evident under both non-insulin- and insulin-stimulated states, suggesting that FKBP51 acts independent of insulin to improve glucose uptake and whole body glucose homeostasis. Follow-up studies should address whether FKBP51 is involved in insulin-independent glucose uptake through the regulation of auxiliary pathways. For example, AMP-activated protein kinase (AMPK) is a well-known regulator of insulin-independent glucose uptake, leading to increased AS160 phosphorylation and GLUT4 translocation in the skeletal muscle[45, 46]. Furthermore, glucocorticoids are potent regulators of glucose homeostasis, and they have been shown to reduce insulin-stimulated glucose transport in muscle by blocking the recruitment of GLUT4 to the cell surface[47]. Indeed, FKBP51 is known to reduce glucocorticoid receptor sensitivity[1], and therefore glucocorticoid signaling is another strong candidate pathway by which FKBP51 may regulate whole body glucose homeostasis. The data herein nevertheless provide unequivocal evidence that FKBP51 is a novel regulator of AKT2-AS160 signaling and glucose uptake.

Not only did FKBP51 antagonism parallel the metabolic phenotype arising from FKBP51 deletion, but it furthermore paralleled the molecular events induced by FKBP51 loss. SAFit2 treatment strongly induced AKT2 and AS160 phosphorylation and led to increased GLUT4 expression at the plasma membrane. Although we suspect that enhanced plasma membrane GLUT4 expression arises from pAKT2-pAS160-mediated increased GLUT4 translocation to the membrane, it is possible that SAFit2 treatment also increases the total expression level of GLUT4. At a functional level, SAFit2 heightened glucose uptake assessed in primary muscle cells. Importantly, SAFit2 action on glucose tolerance and glucose uptake was highly specific for FKBP51 since neither 51KO mice nor primary muscle cells collected from 51KO mice responded to SAFit2 treatment. The robust effects of FKBP51 antagonist SAFit2 on whole body glucose homeostasis and skeletal muscle glucose uptake led us to examine whether SAFit2 disrupts the well-characterized interaction between FKBP51, AKT2, and PHLPP1[16]. To our surprise we found no effect of SAFit2. Rather, we discovered a novel interaction between FKBP51 and AS160, which can be disrupted by FKBP51

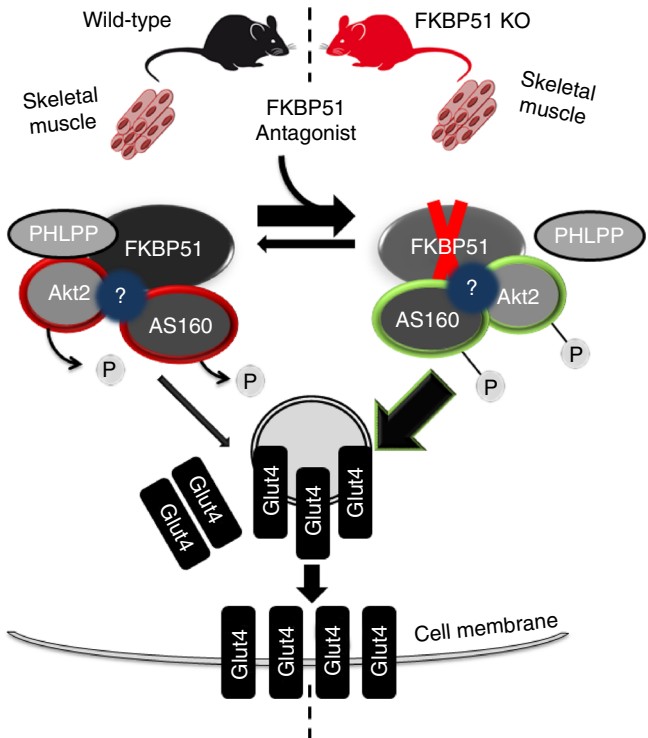

**Fig. 7** Proposed model of FKBP51 as a regulator of glucose uptake. FKBP51 scaffolds Akt2, PHLPP1, and AS160. The associations between FKBP51, AKT2, PHLPP1, and AS160 may be either direct or indirect through additional intermediate proteins. In the presence of FKBP51, PHLPP1 phosphatase activity is directed towards AKT2 to favor inactive Akt2, decreased AS160 phosphorylation and reduced glucose uptake. By contrast, loss of FKBP51's scaffolding function leads to increased AKT2. In the presence of SAFit2, a conformational change within FKBP51 disrupts its ability to form a complex with AS160, while simultaneously enhancing AKT2-AS160 binding. Ultimately, loss of FKBP51 and FKBP51 antagonism with SAFit2 both promote glucose uptake. "?" refers to possible unidentified intermediate proteins within the FKBP51 signaling complex. Curved arrows indicate PHLPP1-mediated dephosphorylation of AKT2 at Ser473. The green double arrow indicates enhanced binding between AKT2 and AS160. Green outlines reflect enhanced phosphorylation; red outlines reflect decreased phosphorylation. The width of the arrows correspond to the magnitude of downstream activation

antagonism using SAFit2. This agrees nicely with our findings that SAFit2 treatment increases glucose uptake in primary myotubes and improves glucose tolerance in mice. Moreover this supports accumulating evidence that FKBP51 has important scaffolding properties to organize and concentrate various signaling complexes[16, 21, 37].

An important question raised by this study is how exactly FKBP51 acts within distinct cellular/tissue environments to affect whole body energy and glucose homeostasis. Previous studies have already alluded to the importance of tissue-specific actions of FKBP51[26, 48]. Here we extend our current understanding of the tissue-specific actions of FKBP51 to include FKBP51-dependent regulation of insulin signaling exclusively within skeletal muscle. Despite microarray-based data demonstrating that skeletal muscle shows the second strongest expression profile of FKBP5 across all tissues examined[12, 14], to our knowledge, the current study is the first to define a skeletal muscle-specific role for FKBP51. The distinct expression profiles of FKBP51 and FKBP52 provide an explanation as to why FKBP51 affects glucose uptake exclusively within skeletal muscle. Although FKBP52 shows strong

expression in WAT, it is expressed minimally in skeletal muscle. We found that FKBP52 competes with FKBP51 for binding to AKT2 in WAT, but this competition is minimal in skeletal muscle based on its expression profile. Importantly, although FKBP52 is found in complex with AKT2 and AS160, it is not found to interact with the negative regulator of AKT2, PHLPP1, and thus does not have the same functional implications on downstream AKT2 signaling compared to FKBP51.

Our study focused on the effects of FKBP51 modulation on glucose homeostasis and pAKT2-pAS160 signaling, effects which were independent of body weight. Nevertheless, we also reported that 51KO mice are resistant to diet-induced obesity, present a higher resting metabolic rate, and a slight increase in home-cage activity. In support of these findings, a recent report demonstrated that UCP1 expression is enhanced in select WAT depot in 51KO mice, which contributes to the improved body weight phenotype[15]. Our finding that 51KO mice also presented increased home-cage activity may furthermore underlie the improved body weight phenotype. Although we found no difference in activity-related energy expenditure, the increased activity in 51KO mice suggests that 51KO mice may have increased exercise efficacy. Follow-up studies are needed to address such effects of FKBP51 on exercise efficacy.

Our working model (Fig. 7) is that FKBP51 scaffolds AKT2, PHLPP1, and AS160 (both phosphorylated and unphosphorylated states), thereby enhancing PHLPP1 phosphatase activity towards AKT2 (at Ser473). Accordingly, higher levels of FKBP51, as is observed under HFD conditions, favors PHLPP1-mediated inactivation of AKT2, activation of AS160, and ultimately lower glucose uptake. Loss of FKBP51 results in AKT2 hyperactivation due to the loss of FKBP51's scaffolding function. Interestingly, in the presence of SAFit2, FKBP51 is no longer able to form a complex with AS160; meanwhile, the AKT2-AS160 binding is enhanced. Consequently, SAFit2 treatment favors a steric confirmation that promotes glucose uptake. In summary, we are the first study to describe a mode of action for SAFit2. Here, the ability of SAFit2 to oppose FKBP51-dependent glucose uptake relates to the disruption of the AS160-FKBP51 complex. Future studies are needed to delineate the full extent of SAFit2 action.

Together, this study defines a novel role for FKBP51 in the regulation of glucose uptake and whole body glucose homeostasis. Furthermore, we established a defined molecular target linking the stress system to the metabolic system, such that FKBP5 induction in response to metabolic stress contributes to the metabolic fate of an individual. Our findings suggest that FKBP51-dependent regulation of AKT2-AS160 signaling contribute to improved glucose tolerance. In addition, we report the first long-term treatment study to employ the recently developed FKBP51 antagonist SAFit2. The positive effects of FKBP51 antagonism on glucose tolerance reported here suggest the opportunity to develop FKBP51 antagonists for the clinic, especially for the treatment of stress-related T2D.

## Methods

**Animals and animal housing**. The *Fkbp5* knockout (51KO) mouse line, used in experiments 1, 2, and 3, had been previously generated[49]. C57BL/6 mice were used in experiment 4 for the pharmacological blockade of FKBP51 (Charles River Laboratories, Maastricht, The Netherlands). For all experiments, male mice between 3 and 4 months old were used. During each experiment, the mice were singly housed. The mice were maintained on a 12:12 h light/dark cycle, with controlled temperature (22 +/− 2 °C) and humidity (55 +/− 5%) conditions. The mice received ad libitum access to water and standard lab chow, unless otherwise specified. The experiments were carried out in accordance with the European Communities' Council Directive 2010/63/EU. The protocols were approved to be carried out at the Max Planck Institute of Psychiatry (license holder) by the ethical committee for the Care and Use of Laboratory animals of the Government of Upper Bavaria, Germany.

**Indirect calorimetry and body composition**. The direct effects of FKBP51 deficiency on metabolic parameters were investigated using 51KO ($n = 16$) and wild-type (WT) ($n = 18$) mice. Body composition (fat and lean mass) was assessed using whole body magnetic resonance imaging (Echo-MRI, Houston, TX). Thereafter, the mice were surgically implanted with a telemetric transponder (Respironics, Murrysville, PA) for the measurement of core body temperature. The mice were allowed to recover for approximately 2 weeks before any metabolic recordings were performed. Indirect calorimetry and telemetry were performed on mice under chow conditions (TSE PhenoMaster, TSE Systems, Bad Homburg, Germany). For experimental details, see Supplemental Information. Each genotype group was subsequently divided into a chow diet and high-fat diet (HFD) (58% kcal from fat, D12331, Research Diets, New Brunswick, NJ, USA) group, matched for body weight. Body weight was measured throughout the experiment. After 8 weeks on the respective diets, 51KO and WT mice were killed. Epididymal (e), inguinal (i), and perirenal (p) WAT were collected and weighed; brown adipose tissue (BAT) was collected and weighed.

**Thermoregulation**. Body weight and body composition were examined in 51KO and WT mice ($n = 8$ per genotype) under HFD conditions at 30 °C to minimize the effects of thermal stress. A separate cohort of 51KO and WT mice were exposed to 6 h of cold exposure (4 °C) to assess cold-induced thermoregulation under both control and HFD conditions (see Supplemental Information).

**Pair-feeding**. To assess the contribution of food intake on body weight regulation in 51KO and WT mice, a pair-feeding experiment was performed. For experimental details, refer to Supplemental Information.

**Glucose tolerance and insulin tolerance**. Glucose tolerance and insulin tolerance were investigated in 51KO ($n = 25$) and WT mice ($n = 18$). Briefly, 51KO and WT mice were initially divided into a control diet group and an HFD group, matched for body weight. After 8 weeks on the dietary treatment, the mice were subjected to a glucose tolerance test (GTT). Additionally, blood was collected to assess fasting insulin and glucose-stimulated insulin levels. One week thereafter, an insulin tolerance test (ITT) was performed. See Supplemental Information for more details.

**SAFit2 administration**. To determine whether antagonizing FKBP51 may be an effective anti-obesity and/or diabetic therapeutic strategy, we treated mice with an antagonist of FKBP51, known as SAFit2[20]. SAFit2 or vehicle was administered either acutely as a slow releasing vesicular phospholipid gel (VPG) (2 mg SAFit2 or vehicle) or repeatedly by intraperitoneal (i.p.) injections (20 mg kg$^{-1}$ SAFit2 or vehicle) twice daily. For i.p. injections, SAFit2 was solubilized in vehicle containing 4% ethanol, 5% Tween80, and 5% PEG400 in 0.9% saline. Body weight and food intake were measured daily throughout the treatment periods. VPGs were composed of 50% (m/m) egg-lecithin containing at least 80% phosphatidylcholine (Lipoid E80, Lipoid GmbH, Ludwigshafen, Germany) and 10 mM phosphate buffered saline (PBS), pH 7.4, and were prepared by a dual asymmetric centrifugation technique[50]. SAFit2 was encapsulated in the formulation by a direct incorporation method. See Supplemental Information for more details.

**Acute SAFit2 treatment**. Male C57BL/6 mice were divided into vehicle-treated and SAFit2-treated groups matched for body weight ($n = 12$ per group). On treatment day 1, mice were administered (subcutaneous injection between the shoulders) a slow release-formulated gel containing either SAFit2 or vehicle. After 48 h, a GTT was performed following an overnight fast.

**Sub-chronic SAFit2 treatment**. One day before the treatment period, male C57BL/6 mice were divided into a vehicle-treated group and a SAFit2-treated group matched for body weight ($n = 8$ per group). On treatment day 7, locomotor activity was assessed in the open field test. On treatment day 8, a GTT was performed. SAFit2 levels were assessed in plasma from blood taken at the time of killing. The animals were killed on day 10 following the treatment onset.

**Chronic SAFit2 treatment**. Four weeks before treatment onset, male C57BL/6 mice were divided into a control diet group ($n = 25$) and an HFD group ($n = 25$) matched for body weight. One day before the treatment period, mice of each dietary group were further subdivided into a vehicle-treated group and a SAFit2-treated group matched for body weight. SAFit2 or vehicle were administered twice daily for 30 days. On treatment days 10 and 30, SAFit2 levels were assessed in plasma. The open field, dark–light transition, and elevated plus-maze behavioral tests were performed on treatment days 15, 16, and 17, respectively. The GTT was performed on treatment day 25 and the ITT on treatment day 29. The animals were killed on day 31 following treatment onset; tissues were collected and stored at −80 °C for further analyses.

**Tissue collection**. Mice were anesthetized with isoflurane and immediately killed by decapitation. Basal trunk blood was collected and subsequently processed (plasma was collected and stored at −20 °C). Skeletal muscle (extensor digitorum longus (EDL) and soleus), WAT (iWAT, eWAT, and pWAT), and liver were collected and stored at −80 °C until used.

**Cell lines**. C2C12 myoblasts were maintained in Dulbecco's modified Eagle's medium (DMEM) supplemented with 10% fetal bovine serum and 1x penicillin streptomycin antibiotics at 37 °C in a humidified atmosphere with 5% $CO_2$. Once the cells reached ~90% confluency, $C_2C1_2$ myoblasts were detached from the plate and $2 \times 10^6$ cells were re-suspended in 100 µl of transfection buffer (50 mM HEPES [pH 7.3], 90 mM NaCl, 5 mM KCl, and 0.15 mM $CaCl_2$). A total of 2.5 µg of plasmid DNA was used per transfection. Plasmids expressing AKT2-HA and FKBP51-FLAG or GFP (control) have been described previously[21, 51]. Ectopic overexpression of AKT2 and FKBP51 resulted in a 3.2-fold and 2.8-fold increase in their expression, respectively. Electroporation was performed using the Amaxa Nucleofector system (program #T-032). The cells were re-seeded onto 0.75% gelatin-coated 12-well plates at a density of ~ $10^5$ cells per cm$^2$. Transfected cells were induced to differentiate once they reached 90% confluency. Differentiation was induced by switching the growth medium to DMEM containing 2% horse serum for 3 days.

Primary EDL myotubes were prepared from satellite cells collected from soleus muscle and EDL myofibers of 4- to 8-week old WT and 51KO mice as described previously[52]. Briefly, dissected muscles were washed in warm 1 x PBS and subsequently were digested in Collagenase 1 at 37 °C for 1.5 h. Thereafter, single fibers were washed in DMEM with 1% P/S. After washing, the fibers were transferred to a 60mm plate (coated with 0,75% gelatin) and were allowed to incubate at 37 °C for 3 days in growth medium (DMEM + 20% FBS + 1%P/S). The satellite cells were detached from the plate and were re-plated onto 24-well multi-well dishes. The medium was exchanged every 2–3 days and cells were split at 90–100% confluency. Differentiation was induced by switching the growth medium to DMEM containing 5% horse serum for 5 days.

**Glucose uptake**. Primary EDL myotubes or transfected C2C12 myotubes were used for glucose uptake experiments. For SAFit2 experiments, a toxicity assay was initially performed to determine the appropriate SAFit2 concentration for subsequent experiments. Based on a lethal dose of 15 (LD 15), the cells were incubated with 0.6 µM SAFit2 or DMSO overnight before inducing glucose uptake.

Basal and insulin-stimulated glucose uptake in primary EDL muscle cells and differentiated C2C12 myotubes was examined. Briefly, the cells were serum-starved in low glucose (1000 mg L$^{-1}$) DMEM for 4 h, and then incubated in Krebs–Ringer-HEPES (KRH) buffer (136 mM NaCl, 4.7 mM KCl, 10 mM sodium phosphate buffer, 1 mM $MgSO_4$, 1 mM $CaCl_2$, and 10 mM HEPES, pH 7.4, 0.2% BSA) for 10 min. The cells were stimulated with insulin (100 nM) or left unstimulated for 1 h. Glucose uptake was induced by the addition of KRH buffer containing 100 µM 2-deoxy-D-[1,2-3 H]glucose, 2 µCi ml$^{-1}$ (Perkin Elmer) to each well. After 4 min, the reactions were terminated by washing the cells with ice-cold PSB containing 10 µM cytochalasin B (inhibitor of membrane transporter-dependent glucose transport), and then 2 additional washes with ice-cold 1x PBS. Cells were lysed with 0.1 M NaOH for 30 min, and the incorporated radioactivity was determined by liquid scintillation counting. 2-deoxy-D-[1,2-3 H]glucose uptake was furthermore normalized to total protein content assessed by the BCA assay (BCA Protein Assay Kit, Life Technologies, Darmstadt, Germany).

**Antibodies**. Detailed information on antibodies and dilutions is provided in Supplementary Information.

**GLUT4 membrane localization**. Primary EDL myotubes were exposed to 0.6 µM SAFit2 or DMSO (vehicle) overnight. The following day, cells were serum-starved in low glucose (1000 mg L$^{-1}$) DMEM for 4 h with SAFit2 or DMSO, and were subsequently collected for the rapid preparation of the plasma membrane fraction as described previously[53]. The membrane fraction was used in subsequent Western blot assays for the detection of GLUT4. For quantification, GLUT4 was normalized to both Na,K,ATPase (plasma membrane marker) and normalized total GLUT4, as described previously[54].

For GLUT4 membrane localization in 51KO ($n = 6$) and WT ($n = 5$) mice, the mice were fasted for 6 h. Insulin was injected by i.p. administration (0.70 IU kg$^{-1}$) 5 min before mice were anesthetized with isoflurane, and were immediately killed by decapitation. Tissues were collected and stored at −80 °C until used. 2-way subcellular fractionation was performed as described previously[55].

**Co-immunoprecipitation (coIP)**. Immunoprecipitations of endogenous proteins were performed using protein extracts ($n = 3$ per group) from soleus muscle, EDL, and eWAT of vehicle-treated and SAFit2-treated mice that were fed HFD. The CoIP experiments were performed with beads conjugated with rabbit IgG. Briefly, 500 µg of lysate was incubated overnight with 2 µg of the appropriate IP-antibody (AKT2 (CST, #2964); FKBP5/FKBP51 (Bethyl, A301-430); and FKBP4/FKBP52 (Bethyl, A301-427A) (See Supplementary Table 1) at 4 °C. 20 µL of protein G dynabeads (Invitrogen, 100-03D) was blocked with bovine serum albumin and subsequently added to the lysate-antibody mix and allowed to incubate at 4 °C for 3 h in order to mediate binding between the dynabeads and the antibody-antigen complex of interest. The beads were washed three times with ice-cold PBS. The

protein-antibody complexes were eluted with 60 μL Laemmli loading buffer. Thereafter, the eluate was boiled for 5 min at 95 °C. Then 2–5 μL of each immunoprecipitate reaction product was separated by SDS-PAGE and electro-transferred onto nitrocellulose membranes. For assessing protein complexes, immunoblotting against AKT2, FKBP51, PHLPP1 (Millipore, #07-1341), and AS160 was performed. See Supplemental Information 'Western blot analysis' for details.

**Statistical analysis**. Data were analyzed using IBM SPSS Statistics 18 software (IBM SPSS Statistics, IBM, Chicago, IL, USA). The decomposition of total energy expenditure (TEE) into activity-related energy expenditure (AEE) and resting metabolic rate (RMR) was performed in MATLAB (The MathWorks, Natick, MA, USA) using a custom-designed toolbox graciously provided by JB van Klinken (Leiden University Medical Center, Leiden, The Netherlands). Body weight was included as a covariate in the analyses of energy expenditure[56]. Statistical analyses for all energy expenditure outcome variables, respiratory exchange ratio (RER), home-cage activity, food intake, water intake, and body temperature were performed on 24-hour averages. Statistical significance was set at $p < 0.05$; a statistical tendency was set at $p < 0.1$. For interactions at $p < 0.1$, we also examined lower order main effects. Data are presented as the mean +/− SEM.

**Data availability**. The data herein are available from the corresponding authors upon reasonable request.

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

## Acknowledgements

The authors thank Marc Cox and Dave Smith (University of Texas at El Paso, El Paso, Texas, US) for originally sharing 51KO mice and MEFs and Lisa Tietze and Jose Monteserin (Max Planck Institute of Psychiatry, Munich, Germany) for their excellent technical assistance. N.C.G. was supported by a European Research Council starting grant (grant# 281338, GxE molmech) within the FP7 framework.

## Author contributions

G.B., A.S.H., N.C.G., M.V.S. and T.R.: Conceived the project and designed the experiments. G.B. and M.L.P.: Managed the colony/genotyping. G.B. and A.S.H.: Performed animal experiments and glucose uptake experiments in cell culture. N.C.G. and K.H.: Performed CoIP experiments, western blot, and subcellular fractionation experiments. C. W.M.: Performed calorimetry experiments and analyses. S.K.: Helped with the calorimetry experiment analyses. M.L.P., C.D., S.S., C.L., A.U. and M.T.: Helped perform animal experiments. X.F. and F.H.: Synthesized SAFit2. M.B. and G.W.: Designed and provided the SAFit2 slow release formulation. C.N. and M.U.: Performed LC/MS/MS. G. B.: Wrote initial version of the manuscript. M.-P.P., A.C., M.H.T., T.R., N.C.G. and M.V. S.: Supervised the research and all authors revised the manuscript.

## Additional information

**Competing interests:** The authors declare no competing financial interests.

