## [Peer Review file · Nature Communications]

Reviewers' comments:

Reviewer #1 (Remarks to the Author):

Balsevich et al present an interesting study of whole body KO and of chemical inhibition of the stress-responsive immunophilin FKBP51, strategies that result in improvement of glucose tolerance in control and high fat diet (HFD)-fed mice. The whole-body phenotype is associated with improved glucose and insulin tolerance without changes in circulating insulin levels, consistent with a phenotype in glucose-storing tissues. Based on the known interaction of FKBP51 with Akt as an inhibitor of all the Akt isoforms, and the shown preferential expression of FKBP51 in muscle, the authors explore insulin signals in muscle tissue and cells in culture. Both FKBP51 KO and chemical antagonism with SAFit2 increased phosphorylation of Akt2 and of its substrate AS160, in muscle but not in liver or adipose tissue. To explore the potential underlying mechanism of this beneficial action of the chemical antagonism of FKBP51, the authors show that acute (6h) injection of SAFit2 into mice reduces AS160 coprecipitation with FKBP51 and promotes its coprecipitation with Akt. In C2C12 myotubes, SAFit2 alone increases membrane-associated GLUT4 and glucose uptake; conversely, overexpression of FKBP51 antagonizes the increase in glucose uptake promoted by Akt overexpression. The authors propose an attractive working model whereby FKBP51 acts as a negative regulator of glucose uptake into muscle by preventing association of Akt2 with AS160 that would purportedly promote GLUT4 translocation; FKBP51 KO or antagonism is proposed to allow this association leading to increased glucose uptake.

MAJOR COMMENTS:

This is an interesting study that goes beyond the typical gene KO study cataloguing phenotype under HFD, by exploring a potential underlying mechanism. This is vastly aided by the use of a relatively novel antagonist of FKBP51 and tests in vivo and on a muscle cell line. The study findings are very interesting in the identification of a muscle-specific action of FKBP51 on insulin-linked signals and metabolism, and on the proposal of a molecular action of FKBP51 to inhibit physical coupling of Akt2 to AS160 (based on the coprecipitation of these proteins induced by the antagonist).

In addition, the manuscript is nicely presented and the Discussion is especially well pitched.

Nonetheless, there are important experiments missing that would enable support of the model as proposed:

1. A major question is that the action of FKBP51 KO or antagonism is often shown only in the presence of insulin, or other times it is unclear if there is insulin present. It is never shown, though, what is the action of the different manipulations side by side on saline and insulin-infused mice. This prevents clear understanding of the precise action of the stress-related protein and its antagonism. Specifically, it is necessary to show the status of pAkt and pAS160 in muscles of the FKBP51 KO mouse (or those infused with SAFit2) in parallel to assessing their status after insulin injection. This would enable understanding of whether FKBP51 inhibition has actions of its own or, as proposed, improves insulin action. This is a very important difference that must be addressed.
2. In a similar vein, the effects of manipulating or inhibiting FKBP51 in C2C12 myotubes must be shown simultaneously with and without insulin. The only experiments that show the no insulin condition are the glucose uptake measurements, leaving a big question mark on the effects on insulin signaling per se.
3. The coprecipitation studies shown in Fig. 5E,F are impressive. However, they only show coprecipitation of the unphosphorylated Akt2 and AS160. Were these experiments done in the presence or absence of insulin? Were the coprecipitating proteins phosphorylated? Knowing this would strengthen the proposed mechanism. In fact, because it has never been shown whether

insulin changes the association of Akt2 and AS160 (without FKBP51 manipulation), doing these experiments would be highly advantageous to cement the model and shed light on the mechanistic significance of the coprecipitation.

4. In Fig 4E, what is the effect of FKBP51 overexpression in the absence of Akt2 overexpression? This is an essential result to evaluate the effect of FKBP51. Moreover, the expression levels shown refer to independent expressions of each protein, so those gels refer to glucose uptake experiments different from those shown.

5. The Table of genes whose expression correlates with levels of FKBP51 in humans is not informative as the correlation could be due to coexisting metabolic parameters. It is recommended that this section be omitted as it does not add to the mechanistic analysis that is the core of the study.

ADDITIONAL SPECIFIC COMMENTS:

1. Does the acute, 6h SAFit2 infusion in vivo improve whole body glucose metabolism?

2. It must be verified that the increase in GLUT4 amount in the membranes shown for either muscle or myotubes is really only in plasma membranes and not in the entire membranes, as FKBP51 KO or inhibition might have altered GLUT4 expression or stability and not necessarily its translocation to the membrane. Since the membrane fractionation protocol used has not been validated in this context, it is highly recommended to provide evidence of plasma membrane marker enrichment vis a vis total membranes.

3. C2C12 myotubes typically do not express GLUT4 or express extremely low levels of this transporter. It is highly recommended to provide experimental evidence that the band shown in the small gel slice illustrated is indeed GLUT4, as the literature has repeatedly been confusing on this subject.

4. If the effects of FKBP51 KO or antagonism occur also in the absence of insulin, it would be important to report if they affect AMPK activation. This is especially important in the cell studies where the antagonist directly promotes glucose uptake in the absence of insulin. Please try to provide these results to verify if the antagonist acts on a separate pathway. Related to this, is the phosphorylation site on AS160 measured also a target of AMPK?

5. Page 8, AS160 is NOT activated by phosphorylation, rather the opposite, it is a negative signal in the insulin pathway (acting as a GAP for Rabs) and its phosphorylation by Akt is thought to remove its inhibitory action.

6. Fig 4E, what is the efficiency of transfection of FKBP1 and of Akt2? This is important as the glucose uptake and signalling results shown are done in the entire culture of myotubes.

7. How is it rationalized into the model that SAFit2 has effects on EWAT? (Fig S5), and are these effects on FKBP51 really (see below).

8. There are a number of important errors throughout the manuscript:

-Fig S5: Why does SAFit2 increase FKBP51 levels?? The figure may be erroneously labelled, and should it perhaps be control and HFD instead of vehicle and SAFit2? Or else it needs an explanation.

-Page 6: title: Antagonist, not antagonists.

-Page 6: Do you really mean to say that after 10 days of FKBP51 antagonism there was a lowering of glucose tolerance?? Throughout the manuscript several times this is stated when the figures and the logic of the paper refers to improved glucose tolerance. This must be thoroughly

revised.

-Page 10, do you really mean to say that loss of FKBP21 lowers glucose tolerance, or rather the opposite, that it improves it??

-Page 11, AS160 is not the most distal insulin signal known, it is in fact the Rab proteins targeted by it, which have been identified in muscle and in adipose cells and differ in each case.

-Coprecipitation does not mean direct interaction, the entire model must be adjusted to accommodate for the formation of a complex that may involve intermediate proteins, and the text should be modified accordingly.

- Page 13, last sentence: do you really mean to say that FKBP21 antagonism has positive effects on body weight? The study shows no effects of the antagonist on body weight.

In spite of all these issues that require attention and experimental addition, the study is very interesting and the authors should be commended for providing mechanistic data on the potential form of action of a novel and promising drug.

Reviewer #2 (Remarks to the Author):

This paper reports that FKBP51 ko (51KO) mice are protected from DIO and have increased muscle insulin sensitivity. A FKBP51 modulator, SAFit2, ameliorated DIO and improved GTT in WT mice. The 51KO mechanism appears to be pAS160 activation in muscle but not WAT.

The paper includes a large amount of data and nicely addresses FKBP51 with a ko mouse, antagonist-like drug, and overexpression in cell lines. This clearly highlights a plausible role for FKBP51 in muscle insulin sensitivity.

1. High levels of FKBP51 are in muscle, and the paper concentrates on muscle physiology. FKBP51 is also in adipose (iWAT) at high levels. What is different about muscle and iWAT to explain why iWAT does not show any pAS160 phenotype?

2. The 51KO mice resist DIO and show a 'modest increase in total energy expenditure'. What is known about the mechanism(s)? What tissue is involved. Is there any evidence of beige/BAT activation? At minimum Ucp1 RNA in WAT/BAT should be presented.

3. The demonstration that chronic treatment with SAFit2 reduced weight/adiposity and improve glucose tolerance is exciting. The improved GTT could be due to the reduced weight/adiposity, directly improved insulin sensitivity, or both. Since weight loss is a common, non-specific sign of drug toxicity, the ms would be greatly improved by showing that SAFit2's actions are lost in the 51KO mice. The acute SAFit2 experiments (fig 5a) would also be much better with 51KO controls showing loss of acute SAFit2 effect in the ko.

4. The 20 mg/kg BID dose is relatively high. How was the SAFit2 dose and dosing regimen chosen? What is known about SAFit2 mouse pharmacokinetics? Are there any dose response data? Any experiments addressing the level and duration of target engagement that was achieved?

5. Minor point—suggest mentioning SAFit2 in the abstract.

Reviewer #3 (Remarks to the Author):

Balsevish et al investigate the role of the chaperon protein encoded by the FKBP51 gene in metabolic regulation. Using primarily animal work they provide genetic and pharmacological

evidence to propose the hypothesis that FKBP51 acts as a negative regulator of insulin signaling toward the glucose transporter translocation machinery and glucose uptake. Specifically they suggest that interaction at the level of Akt-TBC1D4 is key to this role of FKBP51. To bring a human angle to these observations the authors provide gene association data reflecting that FKBP51 gene expression associates with a range of insulin responsive genes in blood cells. The authors conclude that FKBP51 holds high therapeutic potential.

I am puzzled by the combination of the animal model data and the human data. If I had seen the human observations I would first thing perform evaluation studies in the genetic and pharmacological models using the same readout. So why is the gene profile and insulin gene expression dependency not evaluated in the model system? Similar, If I had seen the described metabolic phenotype in the model systems I would have looked for human cohorts that could provide association between FKBP51 and these metabolic effects of FKBP51 in human settings. The authors argue that FKBP51 dictates a muscle phenotype – thus, including human muscle data is obvious, and is missing in this MS. So whereas the study brings forward some new possible signaling insights to FKBP51, I think the outcome of both the animal/cell work as well as the human analyses to some extent was predictable from previous observations and further validations are needed. Thus, my enthusiasm for the MS is somewhat dampened.

By KO of FKBP51 a phenotype of dismissed body weight with less fat mass and enhanced lean mass is evident as well as a model resistant to high fat diet induced obesity. This is in good agreement with previous genetic observations showing that FKBP51 SNP is associated with weight loss following Gastric Bypass operation. Observations after prolonged treatment with a recent described inhibitor of FKBP51 reveals a similar phenotype. By IGT the FKBP51 KO mice display a prolonged period of decreased plasma glucose compared to WT mice, - the authors interpret this as indices of improved peripheral insulin action. Another view could be less counter regulatory response to hypoglycemia (?). Could this be related to the known effect on the corticoid receptor signaling by FKBP51? Nevertheless, a role of FKBP51 in muscle is suggested by elevated membrane Glut4 and by elevated signaling at Akt473 and TBC1D4 642 phosphorylation (5 min after maximal IGT), two events suggested to be critical for glut4 translocation. Since TBC1D4 is described to be a regulator of Glut4 protein turnover it is critical to show in skeletal muscle i) that total glut4 (and Glut1?) is similar in muscle of WT and KO mice, ii) that the increment in plasma membrane of Glut4 with insulin treatment is different and iii) that the endpoint of glucose transport is affected similarly. In the view of this reviewer the data obtained in C2C12 cells is not worth a lot – one of many reasons being that the expression of Glut4 is extremely low in these cells. Furthermore, in these cells, the ability for insulin to induce glucose uptake is NOT dismissed by FKBP51. In fact, why the authors did chose this inferior model system when they have the KO mice. The observation on Akt473 should be supplemented by T308 phosphorylation as both are regulatory for Akt activity.

Activity related EE in the cage is similar between genotype, but the 51KO animal performed more activity. Does this imply that the KO has increase exercise efficacy? What happens during controlled exercise on a treadmill?

I am not convinced by the data that the whole body metabolic phenotype is unrelated to the elevated lean body mass and the elevated physical activity levels. How is these changes induced? Was the Akt2 OE necessary to seen any effect of SAFit2 in C2C12? What happens in control cells? Page 8 L4 bottom. It is written that the data show that FKBP51 is essential for insulin signaling. In fact, insulin induced signaling/ uptake is normal in the 51KO. Using SAFit2 basal glucose uptake is increased. So in my view it needs to clarify whether the phenotype truly relates to insulin action or to changes in the basal non stimulated muscle/cell.

Minor:

P6 L2 bottom should refer to S5A not 5A

The data given in Fig 5 seemingly suggest that Akt does not associate to FKBP51 in any SAFit2 depending way. I am not sure that the model-figure reflects this?

Reviewers' comments:

Reviewer #1 (expert in glucose uptake and Glut4 translocation)

Balsevich et al present an interesting study of whole body KO and of chemical inhibition of the stress-responsive immunophilin FKBP51, strategies that result in improvement of glucose tolerance in control and high fat diet (HFD)-fed mice. The whole-body phenotype is associated with improved glucose and insulin tolerance without changes in circulating insulin levels, consistent with a phenotype in glucose-storing tissues. Based on the known interaction of FKBP51 with Akt as an inhibitor of all the Akt isoforms, and the shown preferential expression of FKBP51 in muscle, the authors explore insulin signals in muscle tissue and cells in culture. Both FKBP51 KO and chemical antagonism with SAFit2 increased phosphorylation of Akt2 and of its substrate AS160, in muscle but not in liver or adipose tissue. To explore the potential underlying mechanism of this beneficial action of the chemical antagonism of FKBP51, the authors show that acute (6h) injection of SAFit2 into mice reduces AS160 coprecipitation with FKBP51 and promotes its coprecipitation with Akt. In C2C12 myotubes, SAFit2 alone increases membrane-associated GLUT4 and glucose uptake; conversely, overexpression of FKBP51 antagonizes the increase in glucose uptake promoted by Akt overexpression. The authors propose an attractive working model whereby FKBP51 acts as a negative regulator of glucose uptake into muscle by preventing association of Akt2 with AS160 that would purportedly promote GLUT4 translocation; FKBP51 KO or antagonism is proposed to allow this association leading to increased glucose uptake.

MAJOR COMMENTS:

This is an interesting study that goes beyond the typical gene KO study cataloguing phenotype under HFD, by exploring a potential underlying mechanism. This is vastly aided by the use of a relatively novel antagonist of FKBP51 and tests in vivo and on a muscle cell line. The study findings are very interesting in the identification of a muscle-specific action of FKBP51 on insulin-linked signals and metabolism, and on the proposal of a molecular action of FKBP51 to inhibit physical coupling of Akt2 to AS160 (based on the coprecipitation of these proteins induced by the antagonist). In addition, the manuscript is nicely presented and the Discussion is especially well pitched.

Author Response – We thank the reviewer for his or her positive comments and fruitful suggestions.

Nonetheless, there are important experiments missing that would enable support of the model as proposed.

Q1. *A major question is that the action of FKBP51 KO or antagonism is often shown only in the presence of insulin, or other times it is unclear if there is insulin present. It is never shown, though, what is the action of the different manipulations side by side on saline and insulin-infused mice. This prevents clear understanding of the precise action of the stress-related protein and its antagonism. Specifically, it is necessary to show the status of pAkt and pAS160 in muscles of the FKBP51 KO mouse (or those infused with SAFit2) in parallel to assessing their status after insulin injection. This would enable understanding of whether FKBP51 inhibition has actions of its own or, as proposed, improves insulin action. This is a very important difference that must be addressed.*

Author Response - We thank the reviewer for this valuable comment that we failed to address in the earlier edition of the manuscript. In the revised edition of our manuscript, we have completely addressed this ambiguity. Specifically, we assessed the phosphorylation of Akt2 and AS160, GLUT4 translocation, and glucose uptake in both non-insulin-stimulated and insulin-stimulated states. We examined this both *in-vivo* using mice administered (i.p. injections) with saline or insulin (0.7 IU/kg) 5 min before harvesting tissue and in cultured primary muscle cells (treated with or without 100 nM insulin). We found that the effects of FKBP51 are indeed independent of insulin such that FKBP51

inhibition has actions of its own. We have clearly stated this in the results and furthermore discussed the implications of this important distinction in the discussion.

Figure 5. FKBP51 antagonism affects insulin signaling and consequently glucose uptake. (A) The insulin signaling pathway was enhanced in EDL skeletal muscle of mice treated with SAFit2 compared to vehicle-treated mice, independent of insulin, as assessed by pAkt2, and pAS160 protein expression. + significant treatment effect, # significant insulin effect.

Figure S9. Effects of FKBP51 antagonism on insulin signaling pathway and glucose uptake. (A) SAFit2 treatment increases pAkt2 in soleus muscle and insulin increases both pAkt2 and pAS160 in soleus muscle. (B) SAFit2 has no effect on pAkt2 in eWAT, whereas insulin increased pAkt2 expression. + significant treatment effect, # significant insulin effect.

Q2. In a similar vein, the effects of manipulating or inhibiting FKBP51 in C2C12 myotubes must be shown simultaneously with and without insulin. The only experiments that show the no insulin condition are the glucose uptake measurements, leaving a big question mark on the effects on insulin signaling per se.

Author Response - We have significantly extended our *in-vitro* studies, including the glucose uptake experiments, in order to show the effects of FKBP51 deletion and FKBP51 antagonism (using SAFit2) on pAkt2, pAS160, GLUT4 translocation, and glucose uptake in primary muscle cells harvested from WT and 51KO mice. In these sets of experiments, we were mindful of the insulin-stimulated condition and therefore we performed all of our new experiments simultaneously with our without insulin.

Figure 4. FKBP51 affects insulin signaling and consequently glucose uptake. ... (D) In primary EDL myotubes, loss of FKBP51 heightened glucose uptake under both no insulin and insulin-stimulated states. + significant treatment effect, # significant insulin effect.

Figure 5. FKBP51 antagonism affects insulin signaling and consequently glucose uptake. ... (C) GLUT4 expression at the membrane was increased from SAFit2 treatment in primary EDL myotubes from WT mice, whereas GLUT1 expression was unchanged by SAFit2 treatment. (D) FKBP51 antagonism with SAFit2 increased 2-deoxyglucose uptake in primary EDL muscle cells harvested from WT mice independent of insulin condition. White bars depict Vehicle condition; black bars depict SAFit2 condition. + significant treatment effect, # significant insulin effect, T trend for insulin effect.

Figure S9. Effects of FKBP51 antagonism on insulin signaling pathway and glucose uptake. ... (C – D) SAFit2 increased pAkt2 in primary EDL and soleus myotubes from WT mice, and also increased pAS160 in EDL myotubes. (E – F) By contrast, there was no effect of SAFit2 treatment in cells harvested from 51KO skeletal muscle (EDL or soleus). (G) SAFit2 had no effect on GLUT4 plasma membrane expression or (H) glucose uptake in primary EDL muscle cells harvested from 51KO mice. The effects of SAFit2 were independent of insulin condition. White bars depict Vehicle condition; black bars depict SAFit2 condition. + significant treatment effect, # significant insulin effect.

Q3. *The coprecipitation studies shown in Fig. 5E,F are impressive. However, they only show coprecipitation of the unphosphorylated Akt2 and AS160. Were these experiments done in the presence or absence of insulin? Were the coprecipitating proteins phosphorylated? Knowing this would strengthen the proposed mechanism. In fact, because it has never been shown whether insulin changes the association of Akt2 and AS160 (without FKBP51 manipulation), doing these experiments would be highly advantageous to cement the model and shed light on the mechanistic significance of the coprecipitation.*

Author Response – In the revised manuscript, we have addressed this comment by not only looking at unphosphorylated AKT2 and AS160, but also at their phosphorylated states. We confirmed that the phosphorylation state of AKT2 and AS160 does not interfere with the interaction between FKBP51 and

either AKT2 or AS160 by examining pAKT2 and pAS160 Fig 5E – F, below). The CoIP experiments were all performed without insulin stimulation.

Figure 5. FKBP51 antagonism affects insulin signaling and consequently glucose uptake. (E – F) Tissue lysates from 30-day vehicle-treated or SAFit2-treated mice exposed to HFD were immunoprecipitated with anti-AKT2 and anti-FKBP51 and then analyzed by Western blot using FKBP51, (p)AKT2, (p)AS160, and PHLPP1. Immunoprecipitation reactions revealed that SAFit2 treatment increased binding between (p)AKT2 and (p)AS160 in soleus (E) and EDL (F) muscles, while simultaneously decreased binding between FKBP51 and AS160 in both muscle types.

Q4. In Fig 4E, what is the effect of FKBP51 overexpression in the absence of Akt2 overexpression? This is an essential result to evaluate the effect of FKBP51. Moreover, the expression levels shown refer to independent expressions of each protein, so those gels refer to glucose uptake experiments different from those shown.

Author Response - We have significantly strengthened our glucose uptake experiments by performing all our revised experiments in primary muscle cells harvested from WT and 51KO mice. This design allows us to see the effects FKBP51 deletion on glucose uptake which aligns more closely to our 51KO mouse model. Here, we demonstrate that primary EDL muscle cells from 51KO mice show heightened glucose uptake compared to those from WT mice in both insulin and non-insulin conditions (Fig 4D shown above). Furthermore, we performed glucose uptake with or without SAFit2 in both insulin- and non-insulin-stimulated conditions. Here, SAFit2 increased glucose uptake again independent of insulin stimulation (Fig 5C above). Importantly, the effects of SAFit2 were not observed in primary EDL cells harvested from 51KO mice (Fig S9H above), demonstrating the specificity of SAFit2 action. We have moved the FKBP51 overexpression study to the supplemental figures (Fig S8) as supporting information to our glucose uptake experiments in primary EDL cells.

Q5. The Table of genes whose expression correlates with levels of FKBP51 in humans is not informative as the correlation could be due to coexisting metabolic parameters. It is recommended that this section be omitted as it does not add to the mechanistic analysis that is the core of the study.

Author Response - We agree with the reviewer. In fact we realized also after reading the comments of reviewer #3 that the human results were still too preliminary to be included in the current study, which is much more focused on the mechanism of action of FKBP51. We have therefore removed the human data from the revised manuscript, which we believe is now much more straightforward.

ADDITIONAL SPECIFIC COMMENTS:

Q1. Does the acute, 6h SAFit2 infusion in vivo improve whole body glucose metabolism?

Author Response - We apologize for the ambiguity. No, despite the evident molecular changes observed 6 h following SAFit2 administration, we observed no effect on glucose tolerance in mice after SAFit2 treatment. There may be several reasons for the lack of phenotypic effect of SAFit2 despite inducing molecular changes at this early time point. As is often the case, molecular changes precede phenotypic changes, but can nevertheless be an early indication of response and modes of action. Regardless, we have now explicitly stated the lack of phenotypic effect in the results section. Furthermore, we have added an additional acute SAFit2 experiment, in which we used a very recently developed slow release formulated gel for the delivery of SAFit2. Here, we showed that as early as 48 h following SAFit2 administration, glucose tolerance in mice is improved.

Q2. *It must be verified that the increase in GLUT4 amount in the membranes shown for either muscle or myotubes is really only in plasma membranes and not in the entire membranes, as FKBP51 KO or inhibition might have altered GLUT4 expression or stability and not necessarily its translocation to the membrane. Since the membrane fractionation protocol used has not been validated in this context, it is highly recommended to provide evidence of plasma membrane marker enrichment vis a vis total membranes.*

Author Response –We thank the reviewer for this comment. In the revised manuscript, we used a different protocol for the preparation of the plasma membrane fraction, which has been validated for the detection of translocated GLUT4¹. We furthermore normalized GLUT4 expression to both a plasma membrane marker (Na,K,ATPase) and total GLUT4 expression as described previously². We have now also examined GLUT1 expression. Importantly, we performed the set of GLUT4/1 translocation experiments in primary EDL myotubes harvested from WT and 51KO mice. These new experiments support our initial findings in 51KO mice (Fig 4C). We report that FKBP51 antagonism increased GLUT4 expression in the plasma membrane fraction of primary EDL myotubes from WT mice, but not 51KO mice (Fig. 5B & Fig. S9G shown above). By contrast, FKBP51 had no effect on GLUT1 expression at the plasma membrane.

Q3. *C2C12 myotubes typically do not express GLUT4 or express extremely low levels of this transporter. It is highly recommended to provide experimental evidence that the band shown in the small gel slice illustrated is indeed GLUT4, as the literature has repeatedly been confusing on this subject.*

Author Response - We thank the reviewer for this valid concern. In light of concerns raised by the reviewer about the validity of C2C12 myotubes for the investigation of GLUT4 translocation and glucose uptake, we have rather performed new experiments in primary muscle cells harvested either from 51KO or WT mice with SAFit2. Using these cells, we find that SAFit2 increased GLUT4 expression in the plasma membrane fraction of primary EDL myotubes from WT mice, but not 51KO mice as shown above (Fig. 5B & Fig. S9G).

Q4. *If the effects of FKBP51 KO or antagonism occur also in the absence of insulin, it would be important to report if they affect AMPK activation. This is especially important in the cell studies where the antagonist directly promotes glucose uptake in the absence of insulin. Please try to provide these results to verify if the antagonist acts on a separate pathway. Related to this, is the phosphorylation site on AS160 measured also a target of AMPK?*

Author Response - In the revised manuscript, we demonstrate that the effect of 51KO or antagonism does in fact occur in the absence of insulin. Although we agree with the reviewer that the ability of FKBP51 to regulate AMPK activation should be addressed, we have sufficient data to submit our findings regarding AMPK as a follow-up study. Specifically, FKBP51 is a strong binding partner of the positive upstream regulator of AMPK, liver kinase B1 (LKB1). We have extensive molecular data detailing a novel regulatory role for FKBP51 on AMPK signaling. Therefore, we would like to keep these novel findings as an important follow-up study, which is principally molecular in nature. Nevertheless, we understand the importance of AMPK signaling on insulin-independent glucose uptake and therefore have thoroughly addressed this as a possibility in the discussion section.

Q5. Page 8, AS160 is NOT activated by phosphorylation, rather the opposite, it is a negative signal in the insulin pathway (acting as a GAP for Rabs) and its phosphorylation by Akt is thought to remove its inhibitory action.

Author Response - We have fixed this mistake: “This is in line with increased glucose uptake given Akt2 inactivates AS160, which in turn stimulates the translocation of GLUT4 to the plasma membrane.”

Q6. Fig 4E, what is the efficiency of transfection of FKBP1 and of Akt2? This is important as the glucose uptake and signalling results shown are done in the entire culture of myotubes.

Author Response - This figure has now been move to the supplemental figures, and we have now included the efficiency of transfection in the supplemental methods (in the cell lines subsection).

Q7. How is it rationalized into the model that SAFit2 has effects on EWAT? (Fig S5), and are these effects on FKBP51 really (see below).

Author Response - The reviewer raises a valid concern. The earlier edition of the manuscript failed to address how FKBP51 is able to regulate Akt2-AS160 in skeletal muscle but not in WAT depots. In the revised edition, we fully addressed this ambiguity by examining the expression profile of FKBP52, a structural homolog of FKBP51 with opposing functional effects. Here, we report that FKBP51 and FKBP52 display distinct expression profiles within WAT depots and skeletal muscle which can account for the ability of FKBP51 to enhance Akt2-AS160 signaling exclusively in skeletal muscle (Fig S6 and S10). In particular, whereas in WAT depots levels of FKBP52 are either significantly higher or similar to the expression of FKBP51, in both soleus and EDL skeletal muscle, levels of FKBP51 are significantly higher than that of FKBP52. We furthermore strengthen our model by demonstrating that FKBP51 and FKBP52 compete for binding with Akt2. However, only FKBP51, not FKBP52, binds PHLPP1 so that FKBP51 ultimately has the inhibitory effect on Akt2 activity (Fig. S10).

Figure S6. Molecular characterization of FKBP51. (A) FKBP51 protein expression was detectable in eWAT, iWAT, soleus muscle, and EDL muscle, whereas was not detected in liver, kidney, spleen, pancreas, gut, or BAT. For FKBP51 tissue expression, n = 6 per tissue.

Figure S10. FKBP52 expression and function account for muscle-specific effects of FKBP51 loss and antagonism. (A) Quantified FKBP52 protein expression in eWAT, iWAT, soleus muscle, and EDL muscle, $n = 6$ per tissue. (B) Tissue lysates from 30-day vehicle-treated or SAFit2-treated mice exposed to HFD were immunoprecipitated with anti-FKBP52 and then analyzed by Western blot using FKBP52, (p)AKT2, (p)AS160, and PHLPP1. Immunoprecipitation reactions revealed that FKBP52 is in complex with AKT2 and AS160 but not PHLPP1. (C) Ectopic overexpression of FKBP51 in C2C12 myotubes decreased pAKT and pAS160 expression. By contrast, simultaneous ectopic overexpression of FKBP51 and FKBP52 prevented the FKBP51-dependent decreases in expression of pAKT2 and pAS160. Panel A $n = 6$ per tissue; panel B $n = 3$ per group; panel C $n = 4$ per group. Data are expressed as relative fold change compared to control condition. ‘a.U.’ denotes arbitrary units.

The distinct FKBP51 expression pattern in eWAT under control and HFD-fed conditions supports the body of literature that regards HFD exposure as a metabolic stressor³. FKBP51 is one of the most well-recognized stress-responsive genes. Therefore, under basal conditions FKBP51 expression is either very low or undetectable in eWAT. In a state of metabolic stress, levels of FKBP51 are increased.

There are a number of important errors throughout the manuscript:

-Fig S5: Why does SAFit2 increase FKBP51 levels?? The figure may be erroneously labelled, and should it perhaps be control and HFD instead of vehicle and SAFit2? Or else it needs an explanation.

Author Response - The reviewer is correct. We have corrected the erroneous label so that it now reads control and HFD

-Page 6: title: Antagonist, not antagonists.

Author Response - We have made the according change.

-Page 6: Do you really mean to say that after 10 days of FKBP51 antagonism there was a lowering of glucose tolerance?? Throughout the manuscript several times this is stated when the figures and the logic of the paper refers to improved glucose tolerance. This must be thoroughly revised.

Author Response - We have read through the manuscript and have changed “lowered glucose tolerance” to “improve glucose tolerance”

-Page 10, do you really mean to say that loss of FKBP51 lowers glucose tolerance, or rather the opposite, that it improves it??

Author Response - We have now changed the sentence so that it states that FKBP51 improves glucose tolerance.

-Page 11, AS160 is not the most distal insulin signal known, it is in fact the Rab proteins targeted by it, which have been identified in muscle and in adipose cells and differ in each case.

Author Response - We have rephrased this sentence to the following: “Accordingly, in the present study we determined that FKBP51 is also important for glucose disposal through the regulation of Akt2 and downstream AS160 (Akt substrate of 160 kDa), an important signaling protein involved in insulin-stimulated glucose transport in skeletal muscle”

-Coprecipitation does not mean direct interaction, the entire model must be adjusted to accommodate for the formation of a complex that may involve intermediate proteins, and the text should be modified accordingly.

Author Response - We agree with the reviewer. We have changed the wording throughout the manuscript. In particular, we have attempted to state that FKBP51 is in complex with Akt2 and AS160. Furthermore we have explicitly mentioned that the interaction may be either direct or indirect, requiring additional intermediate proteins. This is also now reflected in our working model.

Figure 6. Proposed model of FKBP51 as a regulator of glucose uptake. FKBP51 scaffolds Akt2 and PHLPP1 to favor inactive Akt2, and subsequently reduces activation of AS160. In the presence of SAFit2, a conformational change within FKBP51 disrupts its ability to form a complex with AS160. This interaction between FKBP51 and AS160 may be either direct or indirect through additional intermediate proteins. Ultimately, loss of FKBP51 and FKBP51 antagonism with SAFit2 support a steric confirmation that favor Akt2-AS160 binding and promote glucose uptake. ? refers to possible unidentified intermediate proteins within the FKBP51 signaling complex.

- Page 13, last sentence: do you really mean to say that FKBP51 antagonism has positive effects on body weight? The study shows no effects of the antagonist on body weight.

Author Response - We thank the reviewer for pointing this out. We have rephrased the sentence and we no longer mention body weight.

Reviewer #2 (expert in obesity treatment) (Remarks to the Author):

This paper reports that FKBP51 ko (51KO) mice are protected from DIO and have increased muscle insulin sensitivity. A FKBP51 modulator, SAFit2, ameliorated DIO and improved GTT in WT mice. The 51KO mechanism appears to be pAS160 activation in muscle but not WAT.

The paper includes a large amount of data and nicely addresses FKBP51 with a ko mouse, antagonist-like drug, and overexpression in cell lines. This clearly highlights a plausible role for FKBP51 in muscle insulin sensitivity.

Author Response – We thank the reviewer for his/her supportive comments and helpful suggestions.

Q1. High levels of FKBP51 are in muscle, and the paper concentrates on muscle physiology. FKBP51 is also in adipose (iWAT) at high levels. What is different about muscle and iWAT to explain why iWAT does not show any pAS160 phenotype?

Author Response - The reviewer raises a valid concern. The earlier edition of the manuscript failed to address how FKBP51 is able to regulate Akt2-AS160 in skeletal muscle but not in WAT depots, despite being expressed in selective WAT depots. In the revised edition, we fully addressed this ambiguity by examining the expression profile of FKBP52, a structural homolog of FKBP51 with opposing functional effects⁴. Here, we report that FKBP51 and FKBP52 display distinct expression profiles within WAT depots and skeletal muscle (Fig S6A and S10A), which can account for the ability of FKBP51 to enhance Akt2-AS160 signaling exclusively in skeletal muscle. In particular, whereas in WAT depots levels of FKBP52 are either significantly higher or similar to the expression of FKBP51, in both soleus and EDL skeletal muscle, levels of FKBP51 are significantly higher than that of FKBP52. We furthermore strengthen our model by demonstrating that FKBP51 and FKBP52 compete for binding to Akt2. However, only FKBP51, not FKBP52, binds PHLPP1 so that FKBP51 ultimately has the inhibitory effect on Akt2 activity (Fig. S10B – C).

Figure S6A. Molecular characterization of FKBP51. (A) FKBP51 protein expression was detectable in eWAT, iWAT, soleus muscle, and EDL muscle, whereas was not detected in liver, kidney, spleen, pancreas, gut, or BAT. For FKBP51 tissue expression, n = 6 per tissue.

Figure S10. FKBP52 expression and function account for muscle-specific effects of FKBP51 loss and antagonism. (A) Quantified FKBP52 protein expression in eWAT, iWAT, soleus muscle, and EDL muscle, $n = 6$ per tissue. (B) Tissue lysates from 30-day vehicle-treated or SAFit2-treated mice exposed to HFD were immunoprecipitated with anti-FKBP52 and then analyzed by Western blot using FKBP52, (p)AKT2, (p)AS160, and PHLPP. Immunoprecipitation reactions revealed that FKBP52 is in complex with AKT2 and AS160 but not PHLPP. (C) Ectopic overexpression of FKBP51 in C2C12 myotubes decreased pAKT and pAS160 expression. By contrast, simultaneous ectopic overexpression of FKBP51 and FKBP52 prevented the FKBP51-dependent decreases in expression of pAKT2 and pAS160. Panel A $n = 6$ per tissue; panel B $n = 3$ per group; panel C $n = 4$ per group. Data are expressed as relative fold change compared to control condition. 'a.U.' denotes arbitrary units.

Q2. The 51KO mice resist DIO and show a ‘modest increase in total energy expenditure’. What is known about the mechanism(s)? What tissue is involved. Is there any evidence of beige/BAT activation? At minimum *Ucp1* RNA in WAT/BAT should be presented.

Author Response - In the revised manuscript, we address this valid concern raised by the reviewer in 2 ways. First, recently published findings have also found that 51KO mice are resistant to HFD-induced weight gain, which was accompanied by an increase in UCP1 expression in WAT depots⁵. We have added this reference to the introduction and furthermore discussed its relevance to our manuscript in the Discussion section. Secondly, we strengthened our earlier findings that the body weight/energy expenditure phenotype is independent of the glucose tolerance phenotype. In the revised manuscript we provide new data, where already 48 hours after a single dose of SAFit2, glucose tolerance is improved in HFD-fed mice without affecting UCP1 expression, energy expenditure, or body weight.

Q3. The demonstration that chronic treatment with SAFit2 reduced weight/adiposity and improve glucose tolerance is exciting. The improved GTT could be due to the reduced weight/adiposity, directly improved insulin sensitivity, or both. Since weight loss is a common, non-specific sign of drug toxicity, the manuscript would be greatly improved by showing that SAFit2's actions are lost in the 51KO mice. The acute SAFit2 experiments (fig 5a) would also be much better with 51KO controls showing loss of acute SAFit2 effect in the ko.

Author Response –We agree with the reviewer that the two control experiments suggested here would strengthen our findings. Therefore, in the revised manuscript we added the necessary control experiments. Specifically, we fed 51KO and WT mice a HFD for 5 weeks before administering the slow release SAFit2 gel, which improved glucose tolerance as early as 48h following administration (this is also new data incorporated into the revised manuscript, Fig 3 A – D, below). We found that glucose tolerance was not improved in 51KO mice (shown in Fig S4B, below), demonstrating the specificity of SAFit2 action. Furthermore, we have performed all glucose uptake experiments in primary muscle cells harvested from either WT (Fig 5 below) or 51KO mice (Fig S9 below). By treating the cells with SAFit2, we can show that the action of SAFit2 is specific for FKBP51 since it has no effect on glucose uptake in 51KO primary muscle cells, whereas greatly ameliorates glucose uptake in WT primary muscle cells.

Figure 3. FKBP51 antagonism parallels the metabolic effects resulting from genetic ablation of FKBP51. (A) A single application of a slow-release formulated SAFit2 gel had no effect on glucose tolerance or (B) body weight under control diet conditions. (C) Under HFD conditions, acute administration of SAFit2 gel still had no effect on body weight but (D) significantly improved glucose tolerance. + significant treatment effect.

Figure S4. In vivo administration of the FKBP51 antagonist SAFit2. ... (B) There was no effect of SAFit2 gel on GTT assessed 48h after administration in 51KO mice fed a HFD.

Figure 5. FKBP51 antagonism affects insulin signaling and consequently glucose uptake. (C) GLUT4 expression at the membrane was increased from SAFit2 treatment in primary EDL myotubes from WT mice, whereas GLUT1 expression was unchanged by SAFit2 treatment. (D) FKBP51 antagonism with SAFit2 increased 2-deoxyglucose uptake in primary EDL muscle cells harvested from WT mice independent of insulin condition. White bars depict Vehicle condition; black bars depict SAFit2 condition. + significant treatment effect, # significant insulin effect, T trend for insulin effect

Figure S9. Effects of FKBP51 antagonism on insulin signaling pathway and glucose uptake... (G) SAFit2 had no effect on GLUT4 plasma membrane expression or (H) glucose uptake in primary EDL muscle cells harvested from 51KO mice. White bars depict Vehicle condition; black bars depict SAFit2 condition. + significant treatment effect, # significant insulin effect.

Although these control experiments do not explicitly demonstrate that the body weight phenotype following 30 days of SAFit2 treatment is not due to drug toxicity, all of our data argue against this. Specifically, food intake and water intake were similar between vehicle- and SAFit2-treated animals, which would normally not be the case from drug toxicity. We measured additional behavioural readouts in order to examine possible unwanted side effects from the 30-day SAFit2 treatment. We did not observe any difference between vehicle-treated and SAFit2-treated mice in the open field as a reflection of locomotor activity, nor in the elevated plus maze or dark-light transition tests, as a reflection of anxiety-like behaviour (Fig S5E-G). Finally, on the day of sacrifice we evaluated the overall tissue health for each animal and we did not see any overt signs of toxicity from SAFit2 treatment.

Q4. *The 20 mg/kg BID dose is relatively high. How was the SAFit2 dose and dosing regimen chosen? What is known about SAFit2 mouse pharmacokinetics? Are there any dose response data? Any experiments addressing the level and duration of target engagement that was achieved?*

The dose was selected based on the effective dose used in acute studies ⁶. Using this dose, we achieved steady-state levels of circulating SAFit2 measured on treatment day 10 and 30 (Fig S5A, below). In the revised manuscript we now also include SAFit2 pharmacokinetics after a single dose of SAFit2 prepared in the new slow release formulation (Fig S4A) used for the 48h treatment experiments. So far we did not test lower doses of SAFit2 to test the minimal effective dose – we now address this limitation in the discussion section.

Figure S4. Acute administration of the FKBP51 antagonist SAFit2. (A) A single application of slow-release formulated SAFit2 gel resulted in high SAFit2 plasma levels measured at 24h, 48h, and 72h post-injection. ...

Figure S5. Repeated administration of the FKBP51 antagonist SAFit2. (A) Administration of SAFit2 (twice daily at 20 mg/kg) by intraperitoneal injections resulted in high SAFit2 plasma levels and minimal inter-animal variability on day 10 and day 30 of treatment schedule

Q5. Minor point—suggest mentioning SAFit2 in the abstract.

Author Response – We have now mentioned SAFit2 in the abstract.

Reviewer #3 (expert in muscle metabolism and insulin signalling) (Remarks to the Author):

Balsevich et al investigate the role of the chaperon protein encoded by the FKBP51 gene in metabolic regulation. Using primarily animal work they provide genetic and pharmacological evidence to propose the hypothesis that FKBP51 acts as a negative regulator of insulin signaling toward the glucose transporter translocation machinery and glucose uptake. Specifically they suggest that interaction at the level of Akt-TBC1D4 is key to this role of FKBP51. To bring a human angle to these observations the authors provide gene association data reflecting that FKBP51 gene expression associates with a range of insulin responsive genes in blood cells. The authors conclude that FKBP51 holds high therapeutic potential.

Q1. *I am puzzled by the combination of the animal model data and the human data. If I had seen the human observations I would first thing perform evaluation studies in the genetic and pharmacological models using the same readout. So why is the gene profile and insulin gene expression dependency not evaluated in the model system? Similar, If I had seen the described metabolic phenotype in the model systems I would have looked for human cohorts that could provide association between FKBP51 and these metabolic effects of FKBP51 in human settings. The authors argue that FKBP51 dictates a muscle phenotype – thus, including human muscle data is obvious, and is missing in this MS. So whereas the study brings forward some new possible signaling insights to FKBP51, I think the outcome of both the animal/cell work as well as the human analyses to some extent was predictable from previous observations and further validations are needed. Thus, my enthusiasm for the MS is somewhat dampened.*

Author Response – We thank the reviewer for this valuable comment. Indeed, we agree that the human data provided in the initial manuscript are too preliminary and does not strengthen our preclinical findings but rather detracts from the main conclusions. Therefore we have removed the human data in the revised manuscript and rather added a substantial amount of new mechanistic data that further support our finding that FKBP51 in muscle shapes glucose tolerance under challenging conditions. This is a novel and in our view very important finding that is of high relevance for basic science and has a high translational potential for the development of FKBP51 antagonists for the treatment of stress-related T2D.

Q2. *By KO of FKBP51 a phenotype of dismissed body weight with less fat mass and enhanced lean mass is evident as well as a model resistant to high fat diet induced obesity. This is in good agreement with previous genetic observations showing that FKBP51 SNP is associated with weight loss following Gastric Bypass operation. Observations after prolong treatment with a recent described inhibitor of FKBP51 reveals a similar phenotype. By IGT the FKBP51 KO mice display a prolonged period of decreased plasma glucose compared to WT mice, - the authors interpret this as indices of improved peripheral insulin action. Another view could be less counter regulatory response to hypoglycemia (?). Could this be related to the known effect on the corticoid receptor signaling by FKBP51?*

Author Response – Indeed FKBP51 is a strong regulator of the glucocorticoid receptor and ultimately the HPA axis. It has been previously reported that FKBP51 deletion and FKBP51 antagonism using SAFit2 both reduce circulating levels of corticosterone in mice^{7,8}. Glucocorticoids are of course potent regulators of glucose homeostasis, and they have been shown to reduce insulin-stimulated glucose transport in muscle by blocking the recruitment of GLUT4 to the cell surface⁹. Therefore, this is indeed a plausible mechanism of FKBP51 action. Our manuscript rather aims to delineate the downstream mechanism of FKBP51 action directed at Akt2-AS160. We provide unequivocal evidence that FKBP51 scaffolds Akt2 and PHLPP1 to favor inactive Akt2, and subsequently reduces activation of AS160. In the presence of SAFit2 (FKBP51 antagonist), a conformational change within FKBP51 disrupts its ability to form a complex with AS160, which furthermore favors Akt2-AS160 binding, representing a steric confirmation that promotes glucose uptake. These mechanistic studies define a novel role of FKBP51 relevant to the regulation of glucose homeostasis. Whether this occurs in the context of glucocorticoid signaling remains an open area of research. Therefore, we agree with the reviewer that the ability of

FKBP51 to regulate the glucocorticoid receptor may be involved in the FKBP51-dependent regulation of glucose homeostasis, and we have now addressed this in the revised Discussion section (page 13, paragraph 2).

Q3. Nevertheless, a role of FKBP51 in muscle is suggested by elevated membrane Glut4 and by elevated signaling at Akt473 and TBC1D4 642 phosphorylation (5 min after maximal IGT), two events suggested to be critical for glut4 translocation. Since TBC1D4 is described to be a regulator of Glut4 protein turnover it is critical to show in skeletal muscle i) that total glut4 (and Glut1?) is similar in muscle of WT and KO mice, ii) that the increment in plasma membrane of Glut4 with insulin treatment is different and iii) that the endpoint of glucose transport is affected similarly.

Author Response – In the revised manuscript, we used a different protocol for the preparation of the plasma membrane fraction, which has been validated for the detection of translocated GLUT4¹. We furthermore normalized GLUT4 expression to both a plasma membrane marker (Na,K,ATPase) and total GLUT4 expression as described previously². We have now also examined GLUT1 expression. Importantly, we performed the set of GLUT4/1 translocation experiments in primary EDL myotubes harvested from WT and 51KO mice. These new experiments support our initial findings in 51KO mice (Fig 4C). We report that FKBP51 antagonism increased GLUT4 expression in the plasma membrane fraction of primary EDL myotubes from WT mice, but not 51KO mice (Fig. 5B & Fig. S9G shown above). By contrast, FKBP51 had no effect on GLUT1 expression at the plasma membrane. Furthermore, there was no genotype effect on total EDL muscle GLUT1 or GLUT4 (Fig S7D).

Figure S7. FKBP51 selectively effects the insulin signaling pathway and glucose uptake. (D) Total GLUT1 and GLUT4 expression EDL muscle was unaffected by FKBP51 deletion. For quantification of phosphorylated protein, n = 4 per genotype. For GLUT1/4 expression in EDL muscle n = 3 per group. Data are expressed as relative fold change compared to wild-type condition. 'n.d.' denotes 'not detectable'

Q4. In the view of this reviewer the data obtained in C2C12 cells is not worth a lot – one of many reasons being that the expression of Glut4 is extremely low in these cells. Furthermore, in these cells, the ability for insulin to induce glucose uptake is NOT dismissed by FKBP51. In fact, why the authors did chose this inferior model system when they have the KO mice. The observation on Akt473 should be supplemented by T308 phosphorylation as both are regulatory for Akt activity.

Author Response –We thank the reviewer for drawing our attention to this concern. In the revised manuscript, we have fully addressed the comment. In particular, we performed all our revised experiments in primary muscle cells harvested from WT and 51KO mice. This design allows us to see the effects FKBP51 deletion on glucose uptake, which aligns more closely to our 51KO mouse model. It indeed represents a superior model system. Specifically, we have harvested primary muscle cells

from 51KO and WT mice in order to assess (i) the phosphorylation of Akt2 and AS160 (WT: Fig S9C – D, 51KO: Fig S9E – F), (ii) GLUT4 translocation (WT: Fig 5B, 51KO: Fig S9G), and (iii) glucose uptake from FKBP51 deletion (Fig 4D) and antagonism (WT: Fig 5C, 51KO: Fig S9H). Using this preferred model system, glucose uptake is enhanced in primary 51KO muscle cells compared to WT cells. Furthermore, in primary muscle cells, SAFit2 (FKBP51 antagonist) enhanced pAkt2, pAS160, GLUT4 translocation, and glucose uptake. Importantly, the SAFit2 action was highly specific for FKBP51 because SAFit2 treatment had no effect in 51KO primary muscle cells.

Figure 4. FKBP51 affects insulin signaling and consequently glucose uptake. ... (D) In primary EDL myotubes, loss of FKBP51 heightened glucose uptake under both no insulin and insulin-stimulated states. + significant treatment effect, # significant insulin effect.

Figure 5. FKBP51 antagonism affects insulin signaling and consequently glucose uptake. (C) GLUT4 expression at the membrane was increased from SAFit2 treatment in primary EDL myotubes from WT mice, whereas GLUT1 expression was unchanged by SAFit2 treatment. (D) FKBP51 antagonism with SAFit2 increased 2-deoxyglucose uptake in primary EDL muscle cells harvested from WT mice independent of insulin condition. White bars depict Vehicle condition; black bars depict SAFit2 condition. + significant treatment effect, # significant insulin effect, T trend for insulin effect

Figure S9. Effects of FKBP51 antagonism on insulin signaling pathway and glucose uptake. ... (C – D) SAFit2 increased pAkt2 in primary EDL and soleus myotubes from WT mice, and also increased pAS160 in EDL myotubes. (E – F) By contrast, there was no effect of SAFit2 treatment in cells harvested from 51KO skeletal muscle (EDL or soleus). (G) SAFit2 had no effect on GLUT4 plasma membrane expression or (H) glucose uptake in primary EDL muscle cells harvested from 51KO mice. The effects of SAFit2 were independent of insulin condition. White bars depict Vehicle condition; black bars depict SAFit2 condition. + significant treatment effect, # significant insulin effect.

We also examined T308 phosphorylation site on AKT2, which was not changed by FKBP51 modulation. This was to be expected because T308 is not affected by PHLPP1, and therefore not by FKBP51 which recruits PHLPP.

Q5. Activity related EE in the cage is similar between genotype, but the 51KO animal performed more activity. Does this imply that the KO has increase exercise efficacy? What happens during controlled exercise on a treadmill?

Author Response—This is an interesting question that was not addressed in our study. We were indeed a bit puzzled by the finding that in the cohort of FKBP51 KO or WT animals tested in metabolic cages 24 hour activity (beam breaks) were slightly elevated in KO mice, given that all other locomotor-

related parameters (in other cohorts of animals) are not significantly different between KO and WT mice. Also in the new cohort of SAFit2-treated mice that were tested in metabolic cages no locomotor effect was found. In the revised manuscript we now strengthened our mechanistic data demonstrating that the effects of FKBP51 on glucose tolerance are independent of the body weight and energy expenditure phenotype. To address the reviewers comment, we have now added in the Discussion section the possibility that the energy expenditure phenotype of FKBP51 KO mice may be due to an increased exercise efficacy.

Q6. *I am not convinced by the data that the whole body metabolic phenotype is unrelated to the elevated lean body mass and the elevated physical activity levels. How is these changes induced?*

Author Response – In the revised manuscript, we address this valid concern raised by the reviewer in 2 ways. First, recently published findings have also found that 51KO mice are resistant to HFD-induced weight gain, which was accompanied by an increase in UCP1 expression in WAT depots⁵. We have added this reference to the introduction and furthermore discussed its relevance to our manuscript in the Discussion section. Secondly, we have evaluated the effects of FKBP51 antagonism using SAFit2 on UCP1 expression. Here, we find that SAFit2 treatment significantly upregulates UCP1 expression specifically in inguinal and perirenal WAT depots. Importantly, 48 h or SAFit2 treatment improved glucose tolerance without affecting UCP1 expression, energy expenditure or body weight, highlighting the independence of both mechanisms.

Q7. *Was the Akt2 OE necessary to seen any effect of SAFit2 in C2C12? What happens in control cells?*

In the revised manuscript we have performed additional experiments to strengthen our SAFit2-mediated effects on glucose uptake. We performed glucose uptake in primary muscle cells harvested from WT and 51KO mice with or without SAFit2 treatment in both insulin- and non-insulin-stimulated conditions. Here, SAFit2 increased glucose uptake independent of insulin stimulation. Importantly, the effects of SAFit2 were not observed in primary EDL cells harvested from 51KO mice, demonstrating the specificity of SAFit2 action. We have moved the FKBP51 overexpression study to the supplemental figures as supporting information to our glucose uptake experiments in primary EDL cells.

Q8. *Page 8 L4 bottom. It is written that the data show that FKBP51 is essential for insulin signaling. In fact, insulin induced signaling/ uptake is normal in the 51KO. Using SAFit2 basal glucose uptake is increased. So in my view it needs to clarify whether the phenotype truly relates to insulin action or to changes in the basal non stimulated muscle/cell.*

Author Response – We thank the reviewer for this valuable comment that we failed to address in the earlier edition of the manuscript. In the revised edition of our manuscript, we have completely addressed this ambiguity. Specifically, we assessed the phosphorylation of Akt2 and AS160, GLUT4 translocation, and glucose uptake in both non-insulin-stimulated and insulin-stimulated states. We examined this both *in vivo* (Fig 5A & Fig S9A – B, below) using mice administered (i.p. injections) with saline or insulin (0.7 IU/kg) 5 min before harvesting tissue and in cultured primary muscle cells (treated with or without 100 nM insulin) (See Figs 4D, 5B – C, S9D- H, above). We found that the effects of FKBP51 are indeed independent of insulin such that FKBP51 inhibition has actions of its own. We have clearly stated this in the results and furthermore discussed the implications of this important distinction in the discussion. We have furthermore revised the wording of the manuscript to accurately depict the role of FKBP51 on glucose uptake.

Figure 5. FKBP51 antagonism affects insulin signaling and consequently glucose uptake. (A) The insulin signaling pathway was enhanced in EDL skeletal muscle of mice treated with SAFit2 compared to vehicle-treated mice, independent of insulin, as assessed by pAkt2, and pAS160 protein expression. White bars depict Vehicle condition; black bars depict SAFit2 condition. + significant treatment effect, # significant insulin effect.

Figure S9. Effects of FKBP51 antagonism on insulin signaling pathway and glucose uptake. (A) SAFit2 treatment increases pAkt2 in soleus muscle and insulin increases both pAkt2 and pAS160 in soleus muscle. (B) SAFit2 has no effect on pAkt2 in eWAT, whereas insulin increased pAkt2 expression. White bars depict Vehicle condition; black bars depict SAFit2 condition. + significant treatment effect, # significant insulin effect.

Minor:

Q1. P6 L2 bottom should refer to S5A not 5A The data given in Fig 5 seemingly suggest that Akt does not associate to FKBP51 in any SAFit2 depending way. I am not sure that the model-figure reflects this?

Author Response – We thank the reviewer for this observation. We have now corrected the mislabeled Figure.

REFERENCES

1. Nishiumi, S. & Ashida, H. Rapid preparation of a plasma membrane fraction from adipocytes and muscle cells: application to detection of translocated glucose transporter 4 on the plasma membrane. *Biosci. Biotechnol. Biochem.* **71**, 2343–6 (2007).
2. Kong, D. *et al.* Overexpression of mitofusin 2 improves translocation of glucose transporter 4 in skeletal muscle of high- fat diet- fed rats through AMP- activated protein kinase signaling. *Mol. Med. Rep.* **8**, 205–10 (2013).
3. Karalis, K. P. *et al.* Mechanisms of obesity and related pathology: linking immune responses to metabolic stress. *FEBS J.* **276**, 5747–54 (2009).
4. Storer, C. L., Dickey, C. A., Galigniana, M. D., Rein, T. & Cox, M. B. FKBP51 and FKBP52 in signaling and disease. *Trends Endocrinol.Metab* **22**, 481–490 (2011).
5. Stechschulte, L. A. *et al.* FKBP51 Null Mice Are Resistant to Diet-Induced Obesity and the PPAR γ Agonist Rosiglitazone. *Endocrinology* **157**, 3888–3900 (2016).
6. Hartmann, J. *et al.* Pharmacological Inhibition of the Psychiatric Risk Factor FKBP51 Has Anxiolytic Properties. *J.Neurosci.* **35**, 9007–9016 (2015).
7. Albu, S. *et al.* Deficiency of FK506-binding protein (FKBP) 51 alters sleep architecture and recovery sleep responses to stress in mice. *J. Sleep Res.* **23**, 176–85 (2014).
8. Gaali, S. *et al.* Selective inhibitors of the FK506-binding protein 51 by induced fit. *Nat. Chem. Biol.* **11**, 33–7 (2015).
9. Weinstein, S. P., Wilson, C. M., Pritsker, A. & Cushman, S. W. Dexamethasone inhibits insulin-stimulated recruitment of {GLUT}4 to the cell surface in rat skeletal muscle. *Metabolism* **47**, 3–6 (1998).

Reviewers' comments:

Reviewer #1 (Remarks to the Author):

Balsevich et al have done a thorough job in answering the comments, and most have been addressed with added and important experimental evidence. The authors did a number of important new experiments with isolated myotubes that are laborious and well carried out.

There are a couple of outstanding issues that they need to correct and others they could hopefully address or refer to:

1. Abstract: do you really mean to say that FKBP51 antagonism decreased the phosphorylation of AS160?? It is the opposite; this continues to show some carelessness in the wording.

2. The question whether insulin changes the association of AS160 and Akt2 was not addressed. The FKBP51 inhibitor used acts through this mechanism, so it is important to know if it is promoting the same mechanism used by insulin. Otherwise, how does insulin act in the untreated state? At the very least comment on this aspect of the model.

3. The question whether FKBP51 inhibition/KO alters total levels of GLUT4 was not addressed, instead ratios of GLUT4 to GLUT1 or Na/K ATPase in the isolated PM were measured. This still leaves the question of whether inhibition of FKBP51 causes actual translocation of GLUT4 or increases the surface amount by virtue of higher levels of net expression. If this information is not available, at least comment on the possible action on GLUT4 net levels.

4. The model needs some clarification: does Akt2 and AS160 association occur in both basal and phosphorylated states? How does this regulate then the function towards downstream action? The diagram needs clarification. The curved arrows in the absence of treatment also need to be clarified. The legend needs to be corrected to indicate phosphorylation rather than activation of AS160 (AS160 is actually inactivated by phosphorylation).

Reviewer #2 (Remarks to the Author):

My main points have been satisfactorily addressed. A few further suggestions:

1. Fig 1a what is 'weight adjusted' lean mass? Explain in legend. If energy expenditure is also normalized to body weight, also explain how and give body weight in the fig legend.

2. Are the GTT AUC units are correct? Should they be (mg/dl)*2h ?

3. Suggest moving Fig S4b,c to fig 3—possibly with analysis by 2-way ANOVA. This is an important control (although the intrinsically better GTT AUC in the ko mice could be masking off-target GTT improvement by drug).

4. Suggest moving fig S9g,h from supplement to main ms.

5. The authors stress the weight-loss independent effects and the role of muscle to the exclusion of WAT. I would suggest a more nuanced presentation. For example, Fig 2e shows a reduced body weight with SAFit2 treatment, which does not reach statistical significance. This could well be due to a modest effect with too little statistical power to detect it (my bet is on this). Also, if a more sensitive measure than body weight, such as fat mass or fat pad weight, had been measured, one might have seen the (expected) change in adiposity. Another example is the browning of WAT in the ko mouse, a known insulin-sensitizing correlate. Is the lack of seeing a (modest) WAT

browning effect of SAFit2 treatment also a sensitivity problem?

Reviewer #3 (Remarks to the Author):

Balsevich and coworkers have done a fine job revising their MS by rephrasing paragraphs and adding new important analyses.

The withdrawal of the human part is on one part understandable but on the other very unfortunate. As stated earlier adding in human data would have increased these pre-clinical data to a whole different level. Hopefully other groups will follow-up on these data. Clearly, without adding such new human data the MS becomes clearer with the present focus on the pre-clinical data.

Exchanging C2C12 with primary myotubes from mice muscle adds only minimal improvement. Of course, now two cell models support the hypothesis. However, even primary myotubes suffer from de-differentiation with severe loss of the muscle phenotype. So, again in my view, - having the mouse and muscle available these analyses could and should have been performed in the mature tissue.

My general view on this study and MS is rather positive, and I congratulate the laboratories with these pre-clinical observations with promising potential. I am sure the data will be inspirational for many. Yet, I must express a little disappointment in that these well funded and renowned laboratories chose not to bring these observations into man. If tissue availability is a key limiting factor and should you feel tempted to challenge/explore the human aspects feel free to contact me.

Reviewers' comments:

Reviewer #1 (Remarks to the Author):

Balsevich et al have done a thorough job in answering the comments, and most have been addressed with added and important experimental evidence. The authors did a number of important new experiments with isolated myotubes that are laborious and well carried out.

There are a couple of outstanding issues that they need to correct and others they could hopefully address or refer to:

Q1. *Abstract: do you really mean to say that FKBP51 antagonism decreased the phosphorylation of AS160?? It is the opposite; this continues to show some carelessness in the wording.*

Author Response – We apologize for this carelessness. We have corrected this oversight in the revised manuscript.

Q2. *The question whether insulin changes the association of AS160 and Akt2 was not addressed. The FKBP51 inhibitor used acts through this mechanism, so it is important to know if it is promoting the same mechanism used by insulin. Otherwise, how does insulin act in the untreated state? At the very least comment on this aspect of the model.*

Author Response

We hope that we have clarified this ambiguity in the revised manuscript. We have now provided more details describing our model (see Q4 below too). Our model is that FKBP51 regulates AKT2-AS160 signaling independent of the insulin state, which we have stated explicitly throughout the manuscript. FKBP51 functions to organize and concentrate signaling complexes. Therefore, the effects of FKBP51 on AKT2-AS160 are dependent on the FKBP51 status, which is sensitive to the environmental status (i.e. stress and high fat diet increase FKBP51 expression). In terms of SAFit2, it disrupts the interaction between FKBP51 and AS160 (independent of insulin), and in doing so strengthens AKT2-AS160. Therefore, insulin acts upstream of FKBP51 at the insulin receptor, and is not itself affected by FKBP51. FKBP51 likewise does not require insulin in order to scaffold the AKT2-AS160 signaling complex.

Q3. *The question whether FKBP51 inhibition/KO alters total levels of GLUT4 was not addressed, instead ratios of GLUT4 to GLUT1 or Na/K ATPase in the isolated PM were measured. This still leaves the question of whether inhibition of FKBP51 causes actual translocation of GLUT4 or increases the surface amount by virtue of higher levels of net expression. If this information is not available, at least comment on the possible action on GLUT4 net levels.*

Author Response: The reviewer is correct to acknowledge this as a valid possibility. In the revised manuscript, we reported our findings as ‘increased GLUT4 expression at the plasma membrane’ instead of reporting ‘increased GLUT4 translocation to the membrane’. Furthermore, we have now discussed this as a possibility in the discussion: ‘SAFit2 treatment strongly induced AKT2 and AS160 phosphorylation and led to increased GLUT4 expression at the plasma membrane. Although we suspect that enhanced plasma membrane GLUT4 expression arises from pAKT2-pAS160-mediated increased GLUT4 translocation to the membrane, it is possible that SAFit2 treatment also increases the total expression level of GLUT4....’

Q4. *The model needs some clarification: does Akt2 and AS160 association occur in both basal and phosphorylated states? How does this regulate then the function towards downstream action? The diagram needs clarification. The curved arrows in the absence of treatment also need to be clarified. The legend needs to be corrected to indicate phosphorylation rather than activation of AS160 (AS160 is*

actually inactivated by phosphorylation).

Author Response We thank the reviewer for his/her attention to detail. We agree that the working model required clarification in order to better reflect our data. We have attempted to provide a more detailed description of the model in the discussion. Furthermore, we have corrected the figure legend and added a few more details to the figure legend and the figure itself.

Reviewer #2 (Remarks to the Author):

My main points have been satisfactorily addressed. A few further suggestions:

Q1. *Fig 1a what is ‘weight adjusted’ lean mass? Explain in legend. If energy expenditure is also normalized to body weight, also explain how and give body weight in the fig legend.*

Author Response In the revised manuscript, we have now clearly stated in the figure legend (Fig. 1) that ‘Lean mass was adjusted for body weight and is expressed for a 30-g mouse.’ With regard to energy expenditure, we treated body weight as a covariate using ANCOVA, which we have stated in the figure legend S1.

Q2. *Are the GTT AUC units are correct? Should they be (mg/dl)*2h?*

Author Response We thank the reviewer for his/her diligence and attention to detail. The reviewer is correct and the units should read ‘((mg/dl)*2h)’. We have corrected this oversight throughout the revised manuscript

Q3. *Suggest moving Fig S4b,c to fig 3—possibly with analysis by 2-way ANOVA. This is an important control (although the intrinsically better GTT AUC in the ko mice could be masking off-target GTT improvement by drug).*

Author Response We thank the reviewer for his/her comment. We have moved FigS4B (GTT in 51KO mice treated with SAFit2) to Fig. 3. We believe Fig S4c (48h energy expenditure from SAFit2 treatment) is better left in the supplements for presentation purposes. We furthermore did not perform a 2-way ANOVA for the GTT using SAFit2 treatment and genotype as independent variables because of our experimental design. First, the cohorts were run independently at two different times. Second, SAFit2 treatment was given to C57BL6 mice and 51KO mice. Although bred on a C57BL6 background, it is preferred to compare genotype differences arising from 51 ablation to their WT littermates.

Q4. *Suggest moving fig S9g,h from supplement to main ms.*

Author Response We agree with the reviewer. Fig S9g,h are very important findings, confirming the specificity of SAFit2 action on PM GLUT4 expression and 2-DG uptake. Therefore, we have added them to the main manuscript as Fig 5D and 5F.

Q5. *The authors stress the weight-loss independent effects and the role of muscle to the exclusion of WAT. I would suggest a more nuanced presentation. For example, Fig 2e shows a reduced body weight with SAFit2 treatment, which does not reach statistical significance. This could well be due to a modest effect with too little statistical power to detect it (my bet is on this). Also, if a more sensitive measure than body weight, such as fat mass or fat pad weight, had been measured, one might have seen the (expected) change in adiposity. Another example is the browning of WAT in the ko mouse, a known insulin-sensitizing correlate. Is the lack of seeing a (modest) WAT browning effect of SAFit2 treatment also a sensitivity problem?*

Author Response We agree that the study may be underpowered in order to detect subtle effects of SAFit2 treatment on body weight or browning of WAT. We have now acknowledged this possibility in the discussion on page 12: ‘. It is possible that our study was underpowered to detect the modest effects of SAFit2 treatment on body weight since the 10-day SAFit2-treated group showed a lower (non-significant) body weight phenotype compared to the vehicle-treated counterparts.’

Reviewer #3 (Remarks to the Author):

Balsevich and coworkers have done a fine job revising their MS by rephrasing paragraphs and adding new important analyses.

Author Response We thank the reviewer for acknowledging our hard work incorporating all of our revisions.

Q1 *The withdrawal of the human part is on one part understandable but on the other very unfortunate. As stated earlier adding in human data would have increased these pre-clinical data to a whole different level. Hopefully other groups will follow-up on these data. Clearly, without adding such new human data the MS becomes clearer with the present focus on the pre-clinical data.*

Author Response We agree that the preclinical data alone present a clearer take-home message, i.e. to highlight the important role of FKBP51 in glucose homeostasis and to delineate the mode of action of SAFit2 in this context. We believe that it is important to publish the preclinical findings, which we trust will spark widespread interest. As a next step, it will indeed be very important to translate these findings to humans, which we certainly plan to do (see also our response to Q3).

Q2 *Exchanging C2C12 with primary myotubes from mice muscle adds only minimal improvement. Of course, now two cell models support the hypothesis. However, even primary myotubes suffer from de-differentiation with severe loss of the muscle phenotype. So, again in my view, - having the mouse and muscle available these analyses could and should have been performed in the mature tissue.*

Author Response We thank the reviewer for his/her comment. Although the mouse is an ideal model to capture the complex, intricate regulation of whole body energy homeostasis, we believe that, for our purposes, primary myotubes are a very strong model. We do characterize the effects of both 51KO and SAFit2 in mice across many metabolic readouts. Our analyses of AKT2-AS160 signaling points to a muscle-specific phenotype, and therefore our follow-up experiments were largely performed in primary myotubes. We took care to perform our experiments in fully differentiated myotubes, which can be visualized in the representative images below (white arrows). Moreover, there are advantages of using primary myotubes over mouse studies. Firstly, we can conclude that the drug acts directly at the skeletal muscle and it is not an indirect effect. Secondly, drug delivery to the cell system shows good penetration, and we do not have to worry about pharmacokinetics.

Figure 1. Isolated primary muscle satellite cells. Satellite cells (black arrows) start to migrate from isolated myotubes after three days in growing medium (A). After 7 days in differentiation medium, primary satellite cells are fully differentiated to primary myotubes (B & C, white arrows).

Q3 *My general view on this study and MS is rather positive, and I congrats the laboratories with these pre-clinical observations with promising potential. I am sure the data will be inspirational for many. Yet, I must express a little disappointment in that these well-funded and renowned laboratories chose not to bring these observations into man. If tissue availability is a key limiting factor and should you feel tempted to challenge/explore the human aspects feel free to contact me.*

Author Response

As indicated above, we fully agree with the reviewer that the comprehensive data set provided in our manuscript will spark a lot of interest and this intriguing mechanism should now be tested in follow-up studies in human tissues and cohorts. Indeed we are exploring this possibility and we would be happy if the reviewer chose to reveal his/her identity and contact us, in order to discuss a potential collaboration on future studies.

REVIEWERS' COMMENTS:

Reviewer #3 (Remarks to the Author):

In general, and as stated earlier. This is a fine Work. We do not share the same view as to the suitability of C2C12 to mimick skeletal muscle. - but this should not limit the possibilities of this manuscript/paper.

I have no futher comments.

NCOMMS-16-14042B Response to Reviewers Letter

REVIEWERS' COMMENTS:

Reviewer #3 (Remarks to the Author):

Comment

In general, and as stated earlier. This is a fine Work. We do not share the same view as to the suitability of C2C12 to mimick skeletal muscle. - but this should not limit the possibilities of this manuscript/paper.

I have no further comments.

Response

We are happy to see that the reviewer agrees that our manuscript/paper should be published in Nature Communications.